# A General Framework for Robust $G$-Invariance in $G$-Equivariant Networks

**Sophia Sanborn**
sanborn@ucsb.edu

**Nina Miolane**
ninamiolane@ucsb.edu

Department of Electrical and Computer Engineering
UC Santa Barbara

## Abstract

We introduce a general method for achieving robust group-invariance in group-equivariant convolutional neural networks ($G$-CNNs), which we call the $G$-triple-correlation ($G$-TC) layer. The approach leverages the theory of the triple-correlation on groups, which is the unique, lowest-degree polynomial invariant map that is also *complete*. Many commonly used invariant maps—such as the `max`—are incomplete: they remove both group and signal structure. A complete invariant, by contrast, removes only the variation due to the actions of the group, while preserving all information about the structure of the signal. The completeness of the triple correlation endows the $G$-TC layer with strong robustness, which can be observed in its resistance to invariance-based adversarial attacks. In addition, we observe that it yields measurable improvements in classification accuracy over standard Max $G$-Pooling in $G$-CNN architectures. We provide a general and efficient implementation of the method for any discretized group, which requires only a table defining the group's product structure. We demonstrate the benefits of this method for $G$-CNNs defined on both commutative and non-commutative groups—$SO(2)$, $O(2)$, $SO(3)$, and $O(3)$ (discretized as the cyclic $C8$, dihedral $D16$, chiral octahedral $O$ and full octahedral $O_h$ groups)—acting on $\mathbb{R}^2$ and $\mathbb{R}^3$ on both $G$-MNIST and $G$-ModelNet10 datasets.

## 1 Introduction

The *pooling* operation is central to the convolutional neural network (CNN). It was originally introduced in the first CNN architecture—Fukushima's 1980 *Neocognitron* [17]—and remained a fixture of the model since. The Neocognitron was directly inspired by the canonical model of the visual cortex as a process of hierarchical feature extraction and local pooling [25, 1]. In both the neuroscience and CNN model, pooling is intended to serve two purposes. First, it facilitates the local-to-global *coarse-graining* of structure in the input. Second, it facilitates *invariance* to local changes—resulting in network activations that remain similar under small perturbations of the input. In this way, CNNs construct hierarchical, multi-scale features that have increasingly large extent and increasing invariance.

The pooling operation in traditional CNNs, typically a local max or average, has remained largely unchanged over the last forty years. The variations that have been proposed in the literature [40, 56] mostly tackle its *coarse-graining* purpose, improve computational efficiency, or reduce overfitting, but do not seek to enhance its properties with respect to *invariance*. Both max and avg operations are reasonable choices to fulfill the goal of coarse-graining within CNNs and $G$-CNNs. However, they are excessively imprecise and lossy with respect to the goal of constructing robust representations of objects that are invariant only to irrelevant visual changes. Indeed, the max and avg operations are invariant to many natural image transformations such as translations and rotations, but also

37th Conference on Neural Information Processing Systems (NeurIPS 2023).

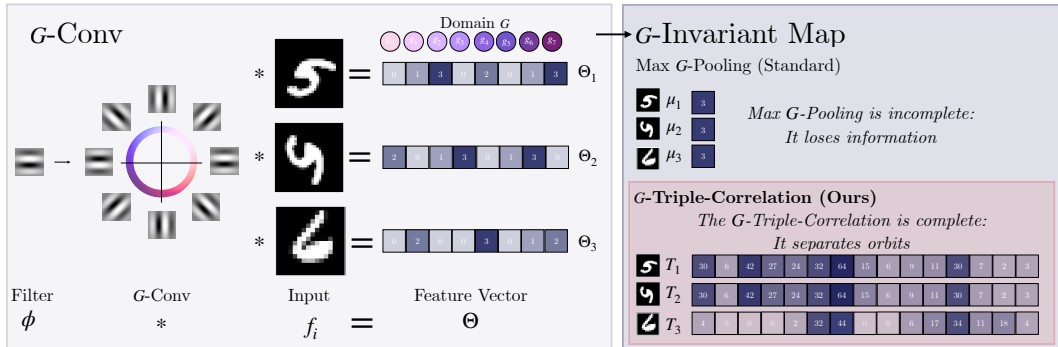

**Figure 1: Achieving Robust $G$-Invariance in $G$-CNNs with the $G$-Triple-Correlation**. The output of a $G$-Convolutional layer is equivariant to the actions of $G$ on the domain of the signal. To identify signals that are equivalent up to group action, the layer can be followed by a $G$-Invariant map that eliminates this equivariance. In $G$-CNNs, Max $G$-Pooling is a commonly used for this purpose. Taking the maximum of the $G$-Convolutional equivariant output is indeed invariant to the actions of the group. However, it is also lossy: many non-equivalent output vectors have the same maximum. Our method—the $G$-*Triple-Correlation* is the lowest-order polynomial invariant map that is *complete* [46]. As a complete invariant, it preserves all information about the signal structure, removing only the action of the group. Our approach thus provides a new foundation for achieving robust $G$-Invariance in $G$-CNNs.

to unnatural transformations, including pixel permutations, that may destroy the image structure. This excessive invariance has been implicated in failure modes such as vulnerability to adversarial perturbations [20, 26], and a bias towards textures rather than objects [4]. To overcome these challenges and enable robust and selective invariant representation learning, there is a need for novel computational primitives that selectively parameterize invariant maps for natural transformations.

Many of the transformations that occur in visual scenes are due to the actions of *groups*. The appreciation of this fact has led to the rise of group-equivariant convolutional networks ($G$-CNNs) [8] and the larger program of Geometric Deep Learning [6]. While this field has leveraged the mathematics of group theory to attain precise generalized group-equivariance in convolutional network layers, the pooling operation has yet to meet its group theoretic grounding. Standardly, invariance to a group $G$ is achieved with a simple generalization of max pooling: Max $G$-Pooling [8] —see Fig. 1 (top-right). However, this approach inevitably suffers from the lossiness of the `max` operation.

Here, we unburden the pooling operation of the dual duty of invariance and coarse-graining, by uncoupling these operations into two steps that can be performed with precision. We retain the standard `max` and `avg` pooling for coarse-graining, but introduce a new method for robust $G$-invariance via the group-invariant triple correlation —see Fig. 1 (bottom-right). The group-invariant triple correlation is the lowest-order complete operator that can achieve exact invariance [32]. As such, we propose a general framework for robust $G$-Invariance in $G$-Equivariant Networks. We show the advantage of this approach over standard max $G$-pooling in several $G$-CNN architectures. Our extensive experiments demonstrate improved scores in classification accuracy in traditional benchmark datasets as well as improved adversarial robustness.

## 2 Background

We first cover the fundamentals of group-equivariant neural networks—also known as $G$-CNNs, or $G$-Equivariant Networks—before introducing the framework for $G$-Invariant Pooling.

### 2.1 Mathematical Prerequisites

The construction of $G$-CNNs requires mathematical prerequisites of group theory, which we recall here. The interested reader can find details in [23].

**Groups.** A *group* $(G, \cdot)$ is a set $G$ with a binary operation $\cdot$, which we can generically call the *product*. The notation $g_1 \cdot g_2$ denotes the product of two elements in the set; however, it is standard to omit

the operator and write simply $g_1 g_2$—a convention we adopt here. Concretely, a group $G$ may define a class of transformations. For example, we can consider the group of two-dimensional rotations in the plane—the special orthogonal group $SO(2)$—or the group of two-dimensional rotations and translations in the plane—the special euclidean group $SE(2)$. Each element of the group $g \in G$ defines a *particular* transformation, such as one *rotation by* $30°$ or one *rotation by* $90°$. The binary operation $\cdot$ provides a means for combining two particular transformations—for example, first rotating by $30°$ and then rotating by $90°$. In mathematics, for a set of transformations $G$ to be a group under the operation $\cdot$, the four axioms of closure, associativity, identity and inverse must hold. These axioms are recalled in Appendix A.

**Group Actions on Spaces.** We detail how a transformation $g$ can transform elements of a space, for example how a rotation of $30°$ indeed rotates a vector in the plane by $30°$. We say that the transformations $g$'s act on (the elements of) a given space. Specifically, consider $X$ a space, such as the plane. A *group action* is a function $L : G \times X \to X$ that maps $(g, x)$ pairs to elements of $X$. We say a group $G$ *acts* on a space $X$ if the following properties of the action $L$ hold:

1. *Identity*: The identity $e$ of the group $G$ "does nothing", i.e., it maps any element $x \in X$ to itself. This can be written as: $L(e, x) = x$.

2. *Compatibility*: Two elements $g_1, g_2 \in G$ can be combined before or after the map $L$ to yield the same result, i.e., $L(g_1, L(g_2, x)) = L(g_1 g_2, x)$. For example, rotating a 2D vector by $30°$ and then $40°$ yields the same result as rotating that vector by $70°$ in one time.

For simplicity, we will use the shortened notation $L_g(x)$ to denote $L(g, x)$ the action of the transformation $g$ on the element $x$.

Some group actions $L$ have additional properties and turn the spaces $X$ on which they operate into *homogeneous spaces*. Homogeneous spaces play an important role in the definition of the $G$-convolution in $G$-CNNs, so that we recall their definition here. We say that $X$ is a *homogeneous space* for a group $G$ if $G$ acts transitively on $X$—that is, if for every pair $x_1, x_2 \in X$ there exists an element of $g \in G$ such that $L_g(x_1) = x_2$. The concept can be clearly illustrated by considering the surface of a sphere, the space $S^2$. The sphere $S^2$ is a homogeneous space for $SO(3)$, the group of orthogonal $3 \times 3$ matrices with determinant one that define 3-dimensional rotations. Indeed, for every pair of points on the sphere, one can define a 3D rotation matrix that takes one to the other.

**Group Actions on Signal Spaces.** We have introduced essential concepts from group theory, where a group $G$ can act on any abstract space $X$. Moving towards building $G$-CNNs, we introduce how groups can act on spaces of signals, such as images. Formally, a *signal* is a map $f : \Omega \to \mathbb{R}^c$, where $\Omega$ is called the domain of the signal and $c$ denotes the number of channels. The *space of signals* itself is denoted $L_2(\Omega, \mathbb{R}^c)$. For example, $\Omega = \mathbb{R}^2$ or $\mathbb{R}^3$ for 2D and 3D images. Gray-scale images have one channel ($c = 1$) and color images have the 3 red-green-blue channels ($c = 3$).

Any action of a group of transformations $G$ on a domain $\Omega$ yields an action of that same group on the spaces of signals defined on that domain, i.e., on $L_2(\Omega, \mathbb{R}^c)$. For example, knowing that the group of 2D rotations $SO(2)$ acts on the plane $\Omega = \mathbb{R}^2$ allows us to define how $SO(2)$ rotates 2D gray-scale images in $L_2(\mathbb{R}^2, \mathbb{R}^c)$. Concretely, the action $L$ of a group $G$ on the domain $\Omega$ yields the following action of $G$ on $L_2(\Omega, \mathbb{R}^c)$:

$$L_g[f](u) = f(L_{g^{-1}}(u)), \qquad \text{for all } u \in \Omega \text{ and for all } g \in G. \tag{1}$$

We use the same notation $L_g$ to refer to the action of the transformation $g$ on either an element $u$ of the domain or on a signal $f$ defined on that domain, distinguishing them using [.] for the signal case. We note that the domain of a signal can be the group itself: $\Omega = G$. In what follows, we will also consider actions on real signals defined on a group, i.e., on signals such as $\Theta : G \to \mathbb{R}$.

**Invariance and Equivariance**. The concepts of group-invariance and equivariance are at the core of what makes the $G$-CNNs desirable for computer vision applications. We recall their definitions here. A function $\psi : X \mapsto Y$ is *$G$-invariant* if $\psi(x) = \psi(L_g(x))$, for all $g \in G$ and $x \in X$. This means that group actions on the input space have no effect on the output. Applied to the group of rotations acting on the space of 2D images $X = L_2(\Omega, \mathbb{R}^c)$ with $\Omega = \mathbb{R}^2$, this means that a $G$-invariant function $\psi$ produces an input that will stay the same for any rotated version of a given signal. For example, whether the image contains the color red is invariant with respect to any rotation of that

image. A function $\psi : X \mapsto Y$ is *G-equivariant* if $\psi(L_g(x)) = L'_g(\psi(x))$ for all $g \in G$ and $x \in X$, where $L$ and $L'$ are two different actions of the group $G$, on the spaces $X$ and $Y$ respectively. This means that a group action on the input space results in a corresponding group action of the same group element $g$ on the output space. For example, consider $\psi$ that represents a neural network performing a foreground-background segmentation of an image. It is desirable for $\psi$ to be equivariant to the group of 2D rotations. This equivariance ensures that, if the input image $f$ is rotated by $30°$, then the output segmentation $\psi(f)$ rotates by $30°$ as well.

## 2.2 $G$-Equivariant Networks

$G$-CNNs are built from the following fundamental building blocks: $G$-convolution, spatial pooling, and $G$-pooling. The $G$-convolution is equivariant to the action of the group $G$, while the $G$-pooling achieves $G$-invariance. Spatial pooling achieves coarse-graining. We review the group-specific operations here. The interested reader can find additional details in [8, 10], which include the definitions of these operations using the group-theoretic framework of principal bundles and associated vector bundles.

### 2.2.1 $G$-Convolution

In plain language, a standard translation-equivariant convolutional neural network layer sweeps filters across a signal (typically, an image), translating the filter and then taking an inner product with the signal to determine the similarity between a local region and the filter. $G$-CNNs [8] generalize this idea, replacing translation with the action of other groups that define symmetries in a machine learning task—for example, rotating a filter, to determine the presence of a feature in various orientations.

Consider a signal $f$ defined on a domain $\Omega$ on which a group $G$ acts. A neural network filter is a map $\phi : \Omega \to \mathbb{R}^c$ defined with the same domain $\Omega$ and codomain $\mathbb{R}^c$ as the signal. A $G$-convolutional layer is defined by a set of filters $\{\phi_1, ..., \phi_K\}$. For a given filter $k$, the layer performs a $G$-*convolution* with the input signal $f$:

$$\Theta_k(g) = (\phi_k * f)(g) = \int_{u \in \Omega} \phi_k(L_{g^{-1}}(u))f(u)du, \quad \forall g \in G, \tag{2}$$

by taking the dot product in $\mathbb{R}^c$ of the signal with a transformed version of the filter. In practice, the domain $\Omega$ of the signal is discretized, such that the $G$-convolutional layer becomes:

$$\Theta_k(g) = \sum_{u \in \Omega} \phi_k(L_{g^{-1}}(u))f(u), \quad \forall g \in G. \tag{3}$$

The output of one filter $k$ is therefore a map $\Theta_k : G \to \mathbb{R}$, while the output of the whole layer with $K$ filters is $\Theta : G \to \mathbb{R}^K$ defined as $\Theta(g) = [\Theta_1(g), \ldots, \Theta_K(g)]$ for all $g \in G$. The $G$-convolution therefore outputs a signal $\Theta$ whose domain has necessarily become the group $\Omega = G$ and whose number of channels is the number of convolutional filters $K$.

The $G$-convolution is *equivariant* to the action of the group on the domain of the signal $f$ [8]. That is, the action of $g$ on the domain of $f$ results in a corresponding action on the output of the layer. Specifically, consider a filter $\phi_k$, we have:

$$\phi_k * L_g[f] = L'_g[\phi_k * f], \qquad \forall g \in G, \tag{4}$$

where $L_g$ and $L'_g$ represent the actions of the same group element $g$ on the functions $f$ and $\phi_k * f$ respectively. This property applies for the $G$-convolutions of the first layer and of the next layers [8].

### 2.2.2 $G$-Pooling

*Invariance* to the action of the group is achieved by pooling over the group ($G$-Pooling) [8]. The pooling operation is typically performed after the $G$-convolution, so that we restrict its definition to signals $\Theta$ defined over a group $G$. In $G$-pooling, a `max` typically is taken over the group elements:

$$\mu_k = \max_{g \in G} \Theta_k(g). \tag{5}$$

$G$-pooling extracts a single real scalar value $\mu_k$ from the full feature vector $\Theta_k$, which has $|G|$ values, with $|G|$ the size of the (discretized) group $G$ as shown in Fig. 1. When the group $G$ is a grid

discretizing $\mathbb{R}^n$, max $G$-Pooling is equivalent to the standard spatial max pooling used in translation-equivariant CNNs, and it can be used to achieve coarse-graining. More generally, $G$-Pooling is $G$-invariant, as shown in [8]. However, we argue here that it is excessively $G$-invariant. Although it achieves the objective of invariance to the group action, it also loses substantial information. As illustrated in Fig. 1, many different signals $\Theta$ may yield same result $\mu$ through the $G$-pooling operation, even if these signals do not share semantic information. This excessive invariance creates an opportunity for adversarial susceptibility. Indeed, inputs $f$ can be designed with the explicit purpose of generating a $\mu_k$ that will fool a neural network and yield an unreasonable classification result. For this reason, we introduce our general framework for robust, selective $G$-invariance.

## 3 The $G$-Triple-Correlation Layer for Robust $G$-Invariance

We propose a $G$-Invariant layer designed for $G$-CNNs that is *complete*—that is, it preserves all information about the input signal except for the group action. Our approach leverages the theory of the triple correlation on groups [32] and applies it to the design of robust neural network architectures. Its theoretical foundations in signal processing and invariant theory allows us to generally define the unique $G$-invariant maps of lowest polynomial order that are complete, hence providing a general framework for selective, robust $G$-invariance in $G$-CNNs [46].

### 3.1 The $G$-Triple-Correlation Layer

The *G-Triple-Correlation* ($G$-TC) on a real signal $\Theta : G \to \mathbb{R}$ is the integral of the signal multiplied by two independently transformed copies of it [32]:

$$\tau_\Theta(g_1, g_2) = \int_{g \in G} \Theta(g)\Theta(gg_1)\,\Theta(gg_2)\,dg. \tag{6}$$

This definition holds for any locally compact group $G$ on which we can define the Haar measure $dg$ used for integration purposes [28]. This definition above is applicable to the $G$-CNNs where $\Theta$ is a collection of scalar signals over the group. We show in Appendix B that we can extend the definition to steerable $G$-CNNs where $\Theta$ can be an arbitrary field [9].

In the equation above, the $G$-TC is computed for a pair of group elements $g_1, g_2$. In practice, we sweep over all pairs in the group. Appendix C illustrates the triple correlation on three concrete groups. Importantly, the $G$-triple-correlation is invariant to the action of the group $G$ on the signal $\Theta$ [28], as shown below.

**Proposition 1.** *Consider a signal $\Theta : G \mapsto \mathbb{R}^c$. The G-Triple-Correlation $\tau$ is G-invariant:*

$$\tau_{L_g[\Theta]} = \tau_\Theta, \quad \text{for all } g \in G, \tag{7}$$

*where $L_g$ denotes an action of a transformation $g$ on the signal $\Theta$.*

The proof is recalled in Appendix D. We propose to achieve $G$-invariance in a $G$-CNN by applying the $G$-Triple-Correlation ($G$-TC) to the output $\Theta$ of a $G$-convolutional layer. Specifically, we apply the $G$-TC to each real scalar valued signal $\Theta_k$ that comes from the $G$-convolution of filter $\phi_k$, for $k \in \{1, ..., K\}$. We only omit the subscript $k$ for clarity of notations. In practice, we will use the triple correlation on discretized groups, where the integral is replaced with a summation:

$$T_\Theta(g_1, g_2) = \sum_{g \in G} \Theta(g)\Theta(gg_1)\Theta(gg_2), \tag{8}$$

for $\Theta$ a scalar valued function defined over $G$. While it seems that the layer computes $T_\Theta(g_1, g_2)$ for all pairs of group elements $(g_1, g_2)$, we note that the real scalars $\Theta(gg_1)$ and $\Theta(gg_2)$ commute so that only half of the pairs are required. We will see that we can reduce the number of computations further when the group $G$ possesses additional properties such as commutativity.

We note that the triple correlation is the spatial dual of the *bispectrum*, which has demonstrated robustness properties in the context of deep learning with bispectral neural networks [42]. The goal of bispectral neural networks is to learn an unknown group $G$ from data. The bispectral layer proposed in [42] assumes an MLP architecture. Our work is the first to generalize the use of bispectral invariants to convolutional networks. Here, we assume that the group $G$ is known in advance, and

exploit the theoretical properties of the triple correlation to achieve robust invariance. One path for future extension may be to combine our approach with the learning approach of [42], to parameterize and learn the group $G$ that defines a $G$-Equivariant and $G$-Invariant layer.

## 3.2 Selective Invariance through Completeness

We show here that the proposed $G$-triple-correlation is guaranteed to preserve all information aside from any equivariant component due to the group action on the input domain. This crucial property distinguishes our proposed layer from standard $G$-Pooling methods, which collapse signals and lose crucial information about the input (Figure 1). In contrast with standard, excessively invariant $G$-pooling methods, we show here that our $G$-TC layer is instead *selectively $G$-invariant* thanks to its *completeness* property [54, 29, 31], defined here:

**Proposition 2.** *Every integrable function with compact support $G$ is completely identified—up to group action—by its $G$-triple-correlation. We say that the $G$-triple-correlation is complete.*

Mathematically, an operator $\mathcal{T}$ is complete for a group action $L$ if the following holds: for every pair of signals $\Theta_1$ and $\Theta_2$, if $\mathcal{T}(\Theta_1) = \mathcal{T}(\Theta_2)$ then the signals are equal up to the group action, that is: there exists a group element $h$ such that $\Theta_2 = L_h[\Theta_1]$.

The proof of the completeness of the $G$-triple-correlation is only valid under a precise set of assumptions [32] (Theorem 2). As we seek to integrate the $G$-triple-correlation to enhance robustness in neural networks, we investigate here the scope of these assumptions. First, the assumptions are not restrictive on the type of groups $G$ that can be used. Indeed, the proof only requires the groups to be Tatsuuma duality groups and the groups of interest in this paper meet this condition. This includes all locally compact commutative groups, all compact groups including the groups of rotations, the special orthogonal groups $SO(n)$, and groups of translations and rotations, the special euclidean groups $SE(n)$. Second, the assumptions are not restrictive on the types of signals. Indeed, the signal only needs to be such that any of its Fourier transform coefficients are invertible. For example, when the Fourier transform coefficients are scalar values, this means that we require these scalars to be non-zero. In practical applications on real image data with noise, there is a probability 0 that the Fourier transform coefficients of the input signal will be exactly 0 (scalar case) or non-invertible (matrix case). This is because the group of invertible matrices is dense in the space of matrices. Therefore, this condition is also verified in the applications of interest and more generally we expect the property of completeness of our $G$-TC layer to hold in practical neural network applications.

## 3.3 Uniqueness

The above two subsections prove that our $G$-Triple Correlation layer is selectively $G$-invariant. Here, we note that our proposed layer is the lowest-degree polynomial layer that can achieve this goal. In invariant theory, it is observed that the $G$-Triple Correlation is the *only* third-order polynomial invariant (up to change of basis) [46]. Moreover, it is the lowest-degree polynomial invariant that is also complete. It thus provides a unique and minimal-complexity solution to the problem of robust invariance within this function class.

## 3.4 Computational Complexity

The $G$-Triple Correlation enjoys some symmetries that we can leverage to avoid computing it for each pair of group elements (which would represent $|G|^2$ computations), hence making the feedforward pass more efficient. We summarize these symmetries here.

**Proposition 3.** *Consider two transformations $g_1, g_2 \in G$. The $G$-Triple Correlation of a real signal $\Theta$ has the following symmetry:*

$$T_\Theta(g_1, g_2) = T_\Theta(g_2, g_1).$$

*If $G$ is commutative, the $G$-Triple Correlation of a real signal has the following additional symmetries:*

$$T_\Theta(g_1, g_2) = T_\Theta(g_1^{-1}, g_2 g_1^{-1}) = T_\Theta(g_2 g_1^{-1}, g_1^{-1}) = T_\Theta(g_2^{-1}, g_1 g_2^{-1}) = T_\Theta(g_1 g_2^{-1}, g_2^{-1}).$$

The proofs are given in [39] for the group of translations. We extend them to any locally compact group $G$ in Appendix E. In practice, these symmetries mean that even if there are theoretically

$|G|^2$ computations, this number immediately reduces to $\frac{|G|(|G|+1)}{2}$ and further reduces if the group $G$ of interest is commutative. In addition, more subtle symmetries can be exploited to reduce the computational cost to linear $|G| + 1$ for the case of one-dimensional cyclic groups [34] by considering the spectral dual of the $G$-TC: the bispectrum. We provide a computational approach to extend this reduction to more general, non-commutative groups in Appendix F. The theory supporting our approach has yet to be extended to this general case. Thus, there is an opportunity for new theoretical work that further increases the computational efficiency of the $G$-Triple-Correlation.

## 4  Related Work

**The Triple Correlation.** The triple correlation has a long history in signal processing [48, 5, 39]. It originally emerged from the study of the higher-order statistics of non-Gaussian random processes, but its invariance properties with respect to translation have been leveraged in texture statistics [53] and data analysis in neuroscience [13], as well as early multi-layer perceptron architectures in the 1990's [12, 33]. The triple correlation was extended to groups beyond translations in [32], and its completeness with respect to general compact groups was established in [30]. To the best of our knowledge, the triple correlation has not previously been introduced as a method for achieving invariance in convolutional networks for either translation or more general groups.

**Pooling in CNNs.** Pooling in CNNs typically has the dual objective of coarse graining and achieving local invariance. While invariance is one desiderata for the pooling mechanism, the machinery of group theory is rarely employed in the computation of the invariant map itself. As noted in the introduction, max and average pooling are by far the most common methods employed in CNNs and $G$-CNNs. However, some approaches beyond strict max and average pooling have been explored. Soft-pooling addresses the lack of smoothness of the max function and uses instead a smooth approximation of it, with methods including polynomial pooling [49] and learned-norm [22], among many others [15, 14, 43, 44, 3, 45, 11, 35]. Stochastic pooling [57] reduces overfitting in CNNs by introducing randomness in the pooling, yielding mixed-pooling [55], max pooling dropout [51], among others [47, 58, 21]

**Geometrically-Aware Pooling.** Some approaches have been adopted to encode spatial or structural information about the feature maps, including spatial pyramid pooling [24], part-based pooling [59], geometric $L_p$ pooling [16] or pooling regions defined as concentric circles [41]. In all of these cases, the pooling computation is still defined by a max. These geometric pooling approaches are reminiscent of the Max $G$-Pooling for $G$-CNNs introduced by [8] and defined in Section 2.2.2, without the explicit use of group theory.

**Higher-Order Pooling.** Average pooling computes first-order statistics (the mean) by pooling from each channel separately and does not account for the interaction between different feature maps coming from different channels. Thus, second-order pooling mechanisms have been proposed to consider correlations between features across channels [38, 19], but higher-orders are not investigated. Our approach computes a third-order polynomial invariant; however, it looks for higher-order correlations within the group rather than across channels and thus treats channels separately. In principle, these approaches could be combined.

## 5  Experiments & Results

### Implementation

We implement the $G$-TC Layer for arbitrary discretized groups with an efficient implementation built on top of the ESCNN library [7, 50], which provides a general implementation of $E(n)$-Equivariant Steerable Convolutional Layers. The method is flexibly defined, requiring the user only to provide a (Cayley) table that defines the group's product structure. The code is publicly available at `https://github.com/sophiaas/gtc-invariance`. Here, we demonstrate the approach on the groups $SO(2)$, and $O(2)$, $SO(3)$, and $O(3)$, discretized as the groups $C_n$ (cyclic), $D_n$ (dihedral), $O$ (chiral octahedral), and $O_h$ (full octahedral), respectively. ESCNN provides implementations for $G$-Conv layers on all of these $E(n)$ subgroups.

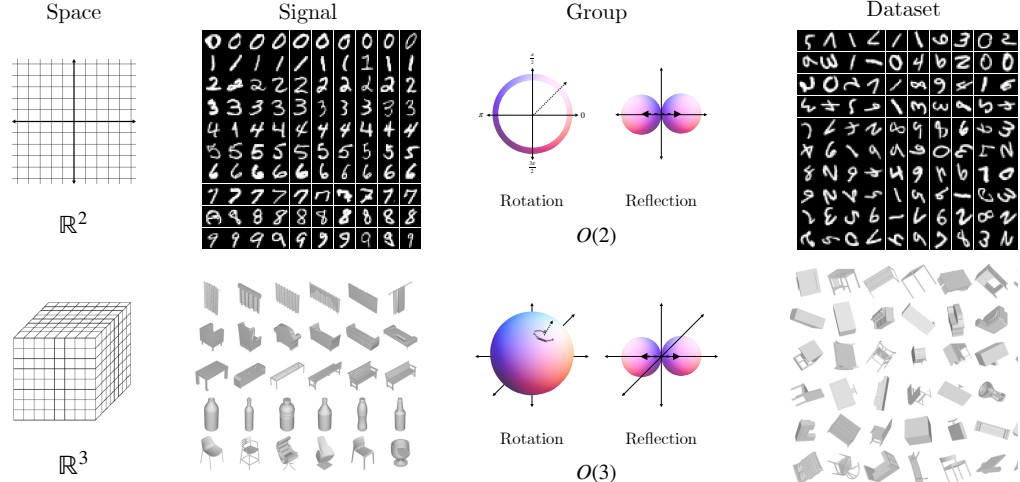

| Space | Signal | Group | Dataset |
| --- | --- | --- | --- |
| $\mathbb{R}^2$ | | Rotation    Reflection
$O(2)$ | |
| $\mathbb{R}^3$ | | Rotation    Reflection
$O(3)$ | |

**Figure 2: Datasets.** The $O(2)$-MNIST (top) and $O(3)$-ModelNet10 (bottom) datasets are generated by applying a random (rotation, reflection) pair to each element of the original datasets. Although we visualize the continuous group here, in practice, we discretize the group $O(3)$ as the full octahedral group $O_h$ to reduce computational complexity. $SO(2)$ and $SO(3)$ datasets are generated similarly, by applying a random rotation to each datapoint.

## Experimental Design

We examine the performance of the $G$-TC over Max $G$-Pooling in $G$-Equivariant Networks defined on these groups and trained on $G$-Invariant classification tasks. For the groups $SO(2)$ and $O(2)$ acting on $\mathbb{R}^2$, we use the MNIST dataset of handwritten characters [37], and for the groups $SO(3)$ and $O(3)$ acting on $\mathbb{R}^3$, we use the voxelized ModelNet10 database of 3D objects [52]. We generate $G$-MNIST and $G$-ModelNet10 datasets by transforming the domain of each signal in the dataset by a randomly sampled group element $g \in G$ (Figure 2).

In these experiments, we train pairs of models in parameter-matched architectures, in which only the $G$-Pooling method differs. Note that the purpose of these experiments is to compare *differences in performance* between models using Max $G$-Pooling vs. the $G$-TC—not to achieve SOTA accuracy. Thus, we do not optimize the models for overall performance. Rather, we fix a simple architecture and set of hyperparameters and examine the change in performance that arises from replacing Max $G$-Pooling with the $G$-TC Layer (Figure 3).

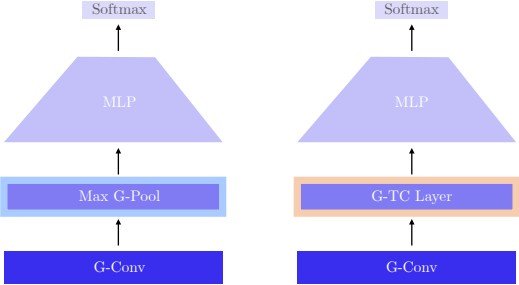

**Figure 3: Models.** We compare two simple architectures comprised of a single G-Conv block followed by either a Max $G$-Pool layer or a $G$-TC Layer and an MLP Classifier.

To isolate the effects of the $G$-Pooling method, all models are comprised of a single $G$-Conv block followed by $G$-Pooling (Max or TC) and an MLP Classifier. Notably, while many $G$-Conv models in the literature use the semi-direct product of $G$ with $\mathbb{R}^n$—i.e. incorporating the actions of the group $G$ into a standard translational convolutional model—here, we perform only *pure $G$-Conv*, without translation. Thus, we use filters the same size as the input in all models. The $G$-Conv block is comprised of a $G$-Conv layer, a batch norm layer, and an optional nonlinearity. For the

Max $G$-Pool model, ReLU is used as the nonlinearity. Given the third-order nonlinearity of the TC, we omit the nonlinearity in the $G$-Conv block in the TC Model. The $G$-TC layer increases the dimensionality of the output of the $G$-Conv block; consequently the input dimension of the first layer of the MLP is larger and the weight matrix contains more parameters than for the Max $G$-Pool model. To compensate for this, we increase the dimension of the output of the first MLP layer in the Max Model, to match the overall number of parameters.

**Evaluation Methods**

We evaluate the models in two ways. First, we examine differences in the raw *classification accuracy* obtained by replacing Max $G$-Pooling with the $G$-TC Layer. Second, we assess the *completeness* of the model by optimizing "metameric" stimuli for the trained models—inputs that yield the same pre-classifier representation as a target input, but are perceptually distinct. The completeness evaluation is inspired by a recent paper that incorporates the bispectrum—the spectral dual of the triple correlation—into a neural network architecture trained to yield $G$-invariant representations for $G$-transformed data [42]. In this work, two inputs are considered "perceptually distinct" if they are not in the same group orbit. They find that all inputs optimized to yield the same representation in the bispectral model are identical up to the group action. By contrast, many metameric stimuli can be found for $E(2)$-CNN [50], a $G$-Equivariant CNN that uses Max $G$-Pooling. Given the duality of the bispectrum and the triple correlation, we expect to observe similar "completeness" for $G$-CNNs using the $G$-TC Layer.

## 5.1 Classification Performance

We train $G$-TC and Max $G$-Pooling models on the $SO(2)$ and $O(2)$-MNIST and chiral ($O$) and full ($O_h$) octahedral voxelized ModelNet10 training datasets and examine their classification performance on the test set. Full training details including hyperparameters are provided in Appendix G. Table 1 shows the test classification accuracy obtained by the Max-$G$ and $G$-TC architectures on each dataset. Accuracy is averaged over four random seeds, with confidence intervals showing standard deviation. We find that the model equipped with $G$-TC obtains a significant improvement in overall classification performance—an increase of 1.3, 0.89, 1.84 and 3.49 percentage points on $SO(2)$-MNIST, $O(2)$-MNIST, $O$-ModelNet10 and $O_h$-ModelNet10 respectively.

| | $C8$-**CNN on** $SO(2)$-**MNIST** | | $D16$-**CNN on** $O(2)$-**MNIST** | |
| --- | --- | --- | --- | --- |
| Method | Accuracy | Parameters | Accuracy | Parameters |
| Max $G$-Pool | $95.23 \pm 0.15$ | 32,915 | $92.17\% \pm 0.23$ | 224,470 |
| $G$-**TC** | $\mathbf{96.53 \pm 0.16}$ | 35,218 | $\mathbf{93.06 \% \pm 0.09}$ | 221,074 |

| | $O$-**CNN on** $O$-**ModelNet10** | | $O_h$-**CNN on** $O_h$-**ModelNet10** | |
| --- | --- | --- | --- | --- |
| Method | Accuracy | Parameters | Accuracy | Parameters |
| Max $G$-Pool | $72.17\% \pm 0.95$ | 500,198 | $71.73\% \pm 0.23$ | 1,826,978 |
| $G$-**TC** | $\mathbf{74.01\% \pm 0.48}$ | $\mathbf{472,066}$ | $\mathbf{75.22\% \pm 0.62}$ | $\mathbf{1,817,602}$ |

**Table 1: Classification Accuracy & Parameter Counts for Models Trained on $G$-MNIST and $G$-ModelNet10**. Confidence intervals reflect standard deviation over four random seeds per model. The model equipped with $G$-TC rather than Max $G$-Pooling obtains significantly improved classification performance on all datasets.

## 5.2 Completeness

Following the analysis of [42], we next evaluate the completeness of the models trained on the $G$-MNIST Dataset. Figure 4 shows inputs optimized to yield the same pre-classifier representation as a set of target images. In line with similar findings from [42], we find that all inputs yielding an identical representations and classifications in the $G$-TC Model are within the same group orbit. Notably, the optimized images are identical to the targets, *up to the group action*. This reflects exactly the completeness of the $G$-TC: the $G$-TC preserves all signal structure up to the group action.

Thus, any rotated version of the a target will yield the same $G$-TC Layer output. By contrast, many "metameric" misclassified stimuli can be found for the Max $G$-Pool Model, a consequence of the lossiness of this pooling operation.

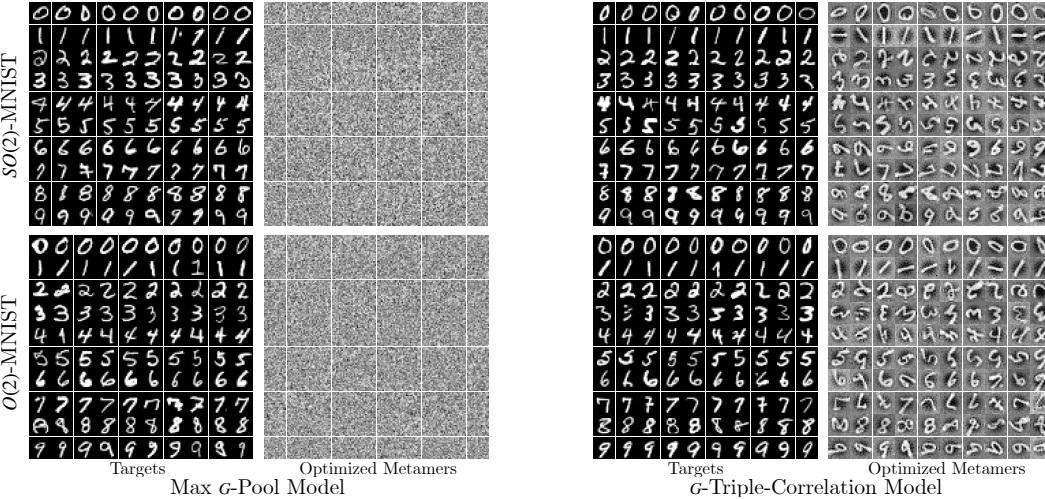

**Figure 4: Optimized Model Metamers.** For each model, 100 targets from the MNIST dataset were randomly selected. 100 inputs were randomly initalized and optimized to yield identical pre-classifier model presentations. All inputs optimized for the $G$-TC Model converge to the orbit of the target. By contrast, metamers that bear no semantic relationship to the targets are found for every target in the Max $G$-Pooling model.

## 6 Discussion

In this work, we introduced a new method for achieving robust group-invariance in group-equivariant convolutional neural networks. Our approach, the *G-TC Layer*, is built on the *triple correlation* on groups, the lowest-degree polynomial that is a complete group-invariant map [32, 46]. Our method inherits its completeness, which provides measurable gains in robustness and classification performance as compared to the ubiquitous Max $G$-Pooling.

This improved robustness comes at a cost: the $G$-TC Layer increases the dimension of the output of a $G$-Convolutional layer from $G$ to $\frac{|G|(|G|+1)}{2}$. While the dimension of the discretized groups used in $G$-CNNs is typically small, this increase in computational cost may nonetheless deter practitioners from its use. However, there is a path to further reduction in computational complexity provided that we consider its spectral dual: the bispectrum. In [34], an algorithm is provided that exploits more subtle symmetries of the bispectrum to demonstrate that only $|G| + 1$ terms are needed to provide a complete signature of signal structure, for the one-dimensional cyclic group. In Appendix F, we extend the computational approach from [34] to more general groups and provided a path for substantial reduction in the complexity of the $G$-TC Layer, thus expanding its practical utility. Novel mathematical work that grounds our proposed computations in group theory is required to quantify the exact complexity reduction that we provide.

As geometric deep learning is applied to increasingly complex data from the natural sciences [18, 2, 27], we expect robustness to play a critical role in its success. Our work is the first to introduce the general group-invariant triple correlation as a new computational primitive for geometric deep learning. We expect the mathematical foundations and experimental successes that we present here to provide a basis for rethinking the problems of invariance and robustness in deep learning architectures.

## Acknowledgments

The authors thank Christopher Hillar, Bruno Olshausen, and Christian Shewmake for many conversations on the bispectrum and triple correlation, which have helped shape the ideas in this work. Thanks also to the members of the UCSB Geometric Intelligence Lab and to four anonymous reviewers for feedback on earlier versions. Lastly, the authors acknowledge financial support from the UC Noyce Initiative: UC Partnerships in Computational Transformation, NIH R01 1R01GM144965-01, and NSF Grant 2134241.

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

# Appendices

## A Group Axioms

For a set of transformations $G$ to be a group under the operation $\cdot$, these four axioms must hold:

1. *Closure*: The product of any two elements of the group is also an element of the group, i.e., for all $g_1, g_2 \in G$, $g_1 g_2 \in G$.
2. *Associativity*: The grouping of elements under the operation does not change the outcome, so long as the order of elements is preserved, i.e., $(g_1 g_2) g_3 = g_1 (g_2 g_3)$.
3. *Identity*: There exists a "do-nothing" *identity* element $e$ that such that the product of $e$ with any other element $g$ returns $g$, i.e., $ge = eg = g$ for all $g \in G$.
4. *Inverse*: For every element $g$, there exists an *inverse* element $g^{-1}$ such that the product of $g$ and $g^{-1}$ returns the identity, i.e. $gg^{-1} = g^{-1}g = e$.

## B The Case of Steerable $G$-CNNs

We consider the framework of Steerable $G$-CNNs defined in [9]. Consider a group $G$ that is the semi-direct product $G = \mathbb{Z}_2 \ltimes H$ of the group of translations $\mathbb{Z}_2$ and a group $H$ of transformations that fixes the origin $0 \in \mathbb{Z}_2$. Consider the feature map $\Theta : \mathbb{Z}_2 \to \mathbb{R}^K$ that is the output of a steerable $G$-CNN. Here, $\Theta$ is a field that transforms according to a representation $\pi$ induced by a representation $\rho$ of $H$ on the fiber $R^K$.

The $G$-TC can be defined on $\Theta$ by replacing the regular representation by the induced representation. Specifically, replace any $\Theta(g)$, i.e. the scalar value of the feature map at group element $g$ by $\pi(g)(\Theta)(x)$, i.e., the vector value of the feature map at $x$ after a group element $g$ has acted on it via the representation $\pi$ :

$$\tau_{\Theta}(g_1, g_2) = \int_G \pi(g)(\Theta)(x)^{\dagger} \cdot \pi(g_1 g)(\Theta)(x) \cdot \pi(g_2 g)(\Theta)(x) dg.$$

Instead of computing the $G$-TC for each scalar coordinate $k$ of $\Theta$ as in the main text, we directly compute it as a vector. The formulation does not depend on the choice of $x$ by homogeneity of the domain $\mathbb{Z}_2$ for the group $G$. Importantly, this $G$-TC is invariant to the action of the induced representation, see Appendix D.

## C The $G$-Triple Correlation: Concrete Examples

We show how to compute the $G$-Triple Correlation ($G$-TC) on three concrete groups. We start with the $G$-TC for the group $\mathbb{R}^2$ of 2D translations.

**Example 1.** *Consider the group of 2D translations $G = (\mathbb{R}^2, +)$ with the addition as the group law. Consider a signal $\Theta$ that is a real function defined over $\mathbb{R}^2$ and can therefore be identified with an image. For any $x_1, x_2 \in \mathbb{R}^2$, the $G$-TC is given by:*

$$\tau_{\Theta}(x_1, x_2) = \int_{x \in \mathbb{R}^2} \Theta(x) \Theta(x + x_1) \Theta(x + x_2) \, dx. \tag{9}$$

Next, we consider the special orthogonal group $SO(2)$ of 2D rotations.

**Example 2.** *Consider the group of 2D rotations $G = (SO(2), \cdot)$ where $SO(2)$ is parameterized by $[0, 2\pi]$ and the composition of rotations $\cdot$ is the addition of angles modulo $2\pi$:*

$$\theta_1 \cdot \theta_2 \equiv \theta_1 + \theta_2 [2\pi]. \tag{10}$$

*For a real signal $\Theta$ defined over $G$, we have:*

$$\tau_{\Theta}(\theta_1, \theta_2) = \int_{\theta \in SO(2)} \Theta(\theta) \Theta(\theta + \theta_1) \Theta(\theta + \theta_2) \, d\theta, \tag{11}$$

*for any $\theta_1, \theta_2 \in SO(2)$ and the addition is taken modulo $2\pi$.*

Finally, we compute the $G$-TC for the special euclidean group $SE(2)$ of 2D rotations and translations, i.e., of 2D rigid-body transformations.

**Example 3.** *Consider the group of 2D rigid body transformations $G = SE(2) = (SO(2) \times \mathbb{R}^2, \cdot)$ equipped with the group composition law:*

$$(\theta_1, x_1) \cdot (\theta_2, x_2) \equiv (\theta_1 + \theta_2, R_{\theta_1}.x_2 + x_1), \tag{12}$$

*where $R_{\theta_1} = \begin{bmatrix} \cos\theta_1 & -\sin\theta_1 \\ \sin\theta_1 & \cos\theta_1 \end{bmatrix}$ is the 2D rotation matrix associated with rotation $\theta_1$ and the addition of angles $\theta_1 + \theta_2$ is taken module $2\pi$.*

*For a real signal defined on $SE(2)$ we have:*

$$\tau_\Theta((\theta_1, x_1), (\theta_2, x_2)) = \int_{(\theta,x) \in SE(2)} \Theta(\theta, x)\Theta\left((\theta, x) \cdot (\theta_1, x_1)\right) \Theta\left((\theta, x) \cdot (\theta_2, x_2)\right) d\theta dx$$

$$= \int_{(\theta,x) \in SE(2)} \Theta(\theta, x)\Theta\left(\theta + \theta_1, R_\theta.x_1 + x\right) \Theta\left(\theta + \theta_2, R_\theta.x_2 + x\right) d\theta dx,$$

*for any $\theta_1, \theta_2 \in SO(2)$ and $x_1, x_2 \in \mathbb{R}^2$.*

## D   Invariance of the $G$-Triple Correlation

Consider a real signal $\Theta$ defined over a group $G$. The $G$-Triple Correlation is invariant to group actions on the domain of the signal $\Theta$ as shown in [28].

**Proposition 4.** *Consider two real signals $\Theta_1, \Theta_2$ defined over a group $G$. If there exists $h \in G$ such that one signal is obtained from the other by a group action, i.e., $\Theta_2 = L_h[\Theta_1]$, then $\tau_{\Theta_1} = \tau_{\Theta_2}$.*

We recall the proof of [28] below.

*Proof.* Consider two real signals $\Theta_1, \Theta_2$ defined over a group $G$, such that $\Theta_2 = L_h[\Theta_1]$ for a group action $L_h$ of group element $h$. We show that this implies that $\tau_{\Theta_1} = \tau_{\Theta_2}$.

Taking $g_1, g_2 \in G$, we have:

$$\tau_{\Theta_2}(g_1, g_2) = \int_{g \in G} \Theta_2(g)\Theta_2(gg_1) \Theta_2(gg_2) dg$$

$$= \int_{g \in G} L_h[\Theta_1](g)L_h[\Theta_1](gg_1) L_h[\Theta_1](gg_2) dg$$

$$= \int_{g \in G} \Theta_1(hg)\Theta_1(hgg_1) \Theta_1(hgg_2) dg$$

$$= \int_{g \in G} \Theta_1(g)\Theta_1(gg_1) \Theta_1(gg_2) dg$$

$$= \tau_{\Theta_1}(g_1, g_2).$$

where we use the change of variable $hg \to g$.

This proves the invariance of the $G$-TC with respect to group actions on the signals.   □

## E   Symmetries of the $G$-Triple Correlation

The $G$-Triple Correlation ($G$-TC) enjoys some symmetries that we can leverage to avoid computing it for each pair of group elements (which would represent $|G|^2$ computations), hence making the feedforward pass more efficient.

These symmetries are given in the main text. We recall them here for completeness.

**Proposition 5.** *Consider two transformations $g_1, g_2 \in G$. The G-Triple Correlation of a signal $\Theta$ has the following symmetry:*

$$(s1) \qquad T_\Theta(g_1, g_2) = T_\Theta(g_2, g_1).$$

*If $G$ is commutative, the G-Triple Correlation of a real signal has the following additional symmetries:*

$$(s2) \qquad T_\Theta(g_1, g_2) = T_\Theta(g_1^{-1}, g_2 g_1^{-1}) = T_\Theta(g_2 g_1^{-1}, g_1^{-1}) = T_\Theta(g_1 g_2^{-1}, g_2^{-1}) = T_\Theta(g_2^{-1}, g_1 g_2^{-1}).$$

Our proof extends the proof given in [39] for the group of translations.

*Proof.* Consider two transformations $g_1, g_2 \in G$.

Symmetry (s1) relies on the fact that $\Theta(gg_2)$ and $\Theta(gg_1)$ are scalar values that commute:

$$T_\Theta(g_2, g_1) = \frac{1}{|G|} \sum_{g \in G} \Theta(g)\Theta(gg_2)\Theta(gg_1) = \frac{1}{|G|} \sum_{g \in G} \Theta(g)\Theta(gg_1)\Theta(gg_2) = T_\Theta(g_2, g_1).$$

For symmetry (s2), we assume that $G$ is commutative. We prove the first equality:

$$
\begin{aligned}
T_\Theta(g_1^{-1}, g_2 g_1^{-1}) &= \frac{1}{|G|} \sum_{g \in G} \Theta(g)\Theta(gg_1^{-1})\Theta(gg_2 g_1^{-1}) \\
&= \frac{1}{|G|} \sum_{g' \in G} \Theta(g' g_1)\Theta(g')\Theta(g' g_1 g_2 g_1^{-1}) \quad \text{(with } g' = gg_1^{-1}, \text{ i.e., } g = g' g_1) \\
&= \frac{1}{|G|} \sum_{g' \in G} \Theta(g' g_1)\Theta(g')\Theta(g' g_2) \quad \text{(} G \text{ commutative: } g_2 g_1^{-1} = g_1^{-1} g_2) \\
&= \frac{1}{|G|} \sum_{g \in G} \Theta(gg_1)\Theta(g)\Theta(gg_2) \\
&= \frac{1}{|G|} \sum_{g \in G} \Theta(g)\Theta(gg_1)\Theta(gg_2) \quad \text{(} \Theta \text{ takes on real values that commute)} \\
&= T_\Theta(g_2, g_1)
\end{aligned}
$$

The second equality of symmetry (s2) follows using (s1). The third and fourth equality of symmetry (s2) have the same proof. $\qquad \square$

This result and its proof are also valid for the extension of the $G$-TC that we propose in Appendix B. They naturally emerge by replacing the regular representation by the induced representation in the proof above.

Specifically, consider a signal $\Theta_2 = \pi(g_0)[\Theta_1]$ obtained from the action of $g_0$ on a signal $\Theta_1$. We want to show that $\tau_{\Theta_1} = \tau_{\Theta_2}$. The key ingredient of the proof is the change of variable within the integral $\int_{G'}$, which follows the semi-direct product structure of G :

$$h' = hh_0 \text{ and } t' = \phi(h)t_0 + t$$

where $\phi(h)$ is a matrix representing h that acts on $\mathbb{Z}_2$ via matrix multiplication: e.g., a rotation matrix $R = \phi(r)$ in the case of $SE(n)$. This concludes the adaptation of the proof for the steerable case.

## F   Algorithmic Reduction

In this section, we show that we can reduce the complexity of the $G$-Triple Correlation of a real signal. This computational reduction requires that we consider, instead, the spectral dual of the $G$-TC, the bispectrum [31]. In what follows, we consider a signal $\Theta$ defined over a finite group $G$. The signal can be real or complex valued.

### F.1   Reduction for Commutative Groups

Consider a commutative group $G$. The bispectrum for a signal $\Theta$ is defined over a pair of irreducible representations $\rho_1, \rho_2$ of the group $G$ as:

$$\beta(\Theta)_{\rho_1, \rho_2} = \mathcal{F}(\Theta)_{\rho_1}^\dagger \mathcal{F}(\Theta)_{\rho_2}^\dagger \mathcal{F}(\Theta)_{\rho_1 \rho_2} \qquad \in \mathbb{C}, \tag{13}$$

where $\mathcal{F}(\Theta)$ is the Fourier transform of $\Theta$ that generalizes the classical Fourier transformation to signals defined over a group. We note that, in group theory, the irreducible representations (irreps) of commutative groups map to scalar values. Hence, the bispectrum is a complex scalar in this case.

For a discrete commutative group, the number of irreps is equal to the size of the group. [**KondorThesis**] proved that, for cyclic groups, it is enough to compute $|G| + 1$ bispectral coefficients to fully describe the signal $\Theta$ up to group action, instead of the $|G|^2$ that would otherwise be required from its definition.

### F.2 Reduction for Non-Commutative Groups

Consider a non-commutative group $G$. The bispectrum of a signal $\Theta$ is defined over a pair of irreducible representations $\rho_1, \rho_2$ of $G$ as:

$$\beta(\Theta)_{\rho_1,\rho_2} = [\mathcal{F}(\Theta)_{\rho_1} \otimes \mathcal{F}(\Theta)_{\rho_2}]^\dagger C_{\rho_1,\rho_2} \Big[ \bigoplus_{\rho \in \rho_1 \otimes \rho_2} \mathcal{F}(\Theta)_\rho \Big] C_{\rho_1,\rho_2}^\dagger \qquad \in \mathbb{C}^{D_1 D_2 \times D_1 D_2},$$

where $\otimes$ is the tensor product, and $\oplus$ is a direct sum over irreps. The unitary Clebsch-Gordan matrix $C_{\rho_1,\rho_2}$ is analytically defined for each pair of representations $\rho_1, \rho_2$ as:

$$(\rho_1 \otimes \rho_2)(g) = C_{\rho_1,\rho_2}^\dagger \Big[ \bigoplus_{\rho \in \rho_1 \otimes \rho_2} \rho(g) \Big] C_{\rho_1,\rho_2}. \tag{14}$$

We note that the irreps of non-commutative groups map to matrices, hence the bispectrum is a complex matrix in this case.

We provide an algorithmic approach reducing the computational complexity of the bispectrum for non-commutative finite groups. We show that we can recover the signal $\Theta$ from a small subset of its bispectral coefficients. That is, we can recover $\Theta$ from coefficients $\beta(\Theta)_{\rho_1,\rho_2}$ computed for a few, well-chosen irreps $\rho_1, \rho_2$. In practice, we only need to compute a few bispectral coefficients to have a complete invariant of the signal $\Theta$ —hence reducing the computational complexity of the layer.

Generalizing [28], we will show that a subset of bispectral coefficients allows us to recover the Fourier transform of the signal $\Theta$ for every irreducible representation $\rho$ of the group. This will show that we can recover the signal $\Theta$ itself by applying the inverse Fourier transform.

We first show relationships between the bispectral coefficients and the Fourier coefficients of the signal $\Theta$ in the following Lemma. We denote $\rho_0$ the trivial representation of the group $G$, which is the representation that sends every group element to the scalar 1.

**Lemma 1.** *Consider $\rho_0$ the trivial representation of the group $G$. Consider $\rho$ any other irreducible representation of dimension $D$. The bispectral coefficients write:*

$$\beta_{\rho_0,\rho_0} = |\mathcal{F}(\Theta)_{\rho_0}|^2 \mathcal{F}(\Theta)_{\rho_0} \in \mathbb{C}$$
$$\beta_{\rho,\rho_0} = \mathcal{F}(\Theta)_{\rho_0}^\dagger \mathcal{F}(\Theta)_\rho^\dagger \mathcal{F}(\Theta)_\rho \in \mathbb{C}^{D \times D}.$$

Here, and in what follows, we denote $\beta$ the bispectrum of $\Theta$, i.e., we omit the argument $\Theta$ for clarity of notations.

*Proof.* For $\rho_0$ the trivial representation, and $\rho$ an irreps, the Clebsh-Jordan (CJ) matrix $C_{\rho\rho_0}$ is the identity matrix, and the matrix $C_{\rho\rho_0}$ is the scalar 1.

We compute the bispectral coefficient $\beta_{\rho_0,\rho_0}$:

$$\beta_{\rho_0,\rho_0} = (\mathcal{F}(\Theta)_{\rho_0} \otimes \mathcal{F}(\Theta)_{\rho_0})^\dagger C_{\rho_0\rho_0} \left[ \bigoplus_{\rho \in \rho_0 \otimes \rho_0} \mathcal{F}(\Theta)_\rho \right] C_{\rho_0\rho_0}^\dagger$$

$$= (\mathcal{F}(\Theta)_{\rho_0} \otimes \mathcal{F}(\Theta)_{\rho_0})^\dagger C_{\rho_0\rho_0} \mathcal{F}(\Theta)_{\rho_0} C_{\rho_0\rho_0}^\dagger \qquad (\rho_0 \otimes \rho_0 = \rho_0 \text{ which is irreducible})$$

$$= (\mathcal{F}(\Theta)_{\rho_0}^2)^\dagger C_{\rho_0\rho_0} \mathcal{F}(\Theta)_{\rho_0} C_{\rho_0\rho_0}^\dagger \qquad (\mathcal{F}(\Theta)_{\rho_0} \text{ is a scalar for which tensor product is multiplication})$$

$$= |\mathcal{F}(\Theta)_{\rho_0}|^2 \mathcal{F}(\Theta)_{\rho_0} \qquad (\text{CJ matrices are equal to 1.})$$

Take any irreducible representation $\rho$ of dimension $D$, we have:

$$\beta_{\rho,\rho_0} = (\mathcal{F}(\Theta)_\rho \otimes \mathcal{F}(\Theta)_{\rho_0})^\dagger C_{\rho\rho_0} \left[ \bigoplus_{\rho \in \rho \otimes \rho_0} \mathcal{F}(\Theta)_\rho \right] C^\dagger_{\rho\rho_0}$$

$$= \mathcal{F}(\Theta)^\dagger_{\rho_0} \mathcal{F}(\Theta)^\dagger_\rho C_{\rho\rho_0} \mathcal{F}(\Theta)_\rho C^\dagger_{\rho\rho_0} \qquad (\mathcal{F}(\Theta)_{\rho_0} \text{ is a scalar and } \rho \otimes \rho_0 = \rho)$$

$$= \mathcal{F}(\Theta)^\dagger_{\rho_0} \mathcal{F}(\Theta)^\dagger_\rho \mathcal{F}(\Theta)_\rho \qquad \text{(CJ matrices are identity matrices)}.$$

$\square$

Next, we summarize our main result.

**Proposition 6.** *We can recover the Fourier coefficients of a signal $\Theta$ from only $L$ bispectral coefficients, where $L$ is a number computed from the Kronecker product table of the group $G$.*

In the proof, we propose an algorithmic method that iteratively computes bispectral coefficients until the Fourier coefficients of the signal are all recovered. We note that, for arbitrary groups and their representations, Clebsch–Gordan (CJ) matrices are not known in general, yet they can be computed numerically. Thus, the proof below assumes that the CJ matrices are given for the group $G$ of interest.

*Proof.* **Algorithmic Approach.**

First, we show how we can recover the first Fourier coefficient (DC component), i.e., the Fourier transform of the signal at the trivial representation $\rho_0$, from a single bispectral coefficient.

$$\mathcal{F}(\Theta)_{\rho_0} = \hat{f}_{\rho_0} = \int_G \Theta(g)\rho_0(g)dg = \int_G \Theta(g)dg \in \mathbb{C}. \tag{15}$$

Using Lemma 1, we can recover this Fourier component from the bispectral coefficient $\beta_{\rho_0,\rho_0}$, as:

$$|\mathcal{F}(\Theta)_{\rho_0}| = (|\beta_{\rho_0,\rho_0}|)^{1/3}, \quad \arg(\mathcal{F}(\Theta)_{\rho_0}) = \arg(\beta_{\rho_0,\rho_0}). \tag{16}$$

Next, consider an irreducible representation $\rho_1$ of dimension $D$. We seek to recover the Fourier coefficient $\mathcal{F}(\Theta)_{\rho_1}$. This Fourier coefficient is a matrix in $\mathbb{C}^{D \times D}$. Using Lemma 1, we can recover it from a single bispectral coefficient:

$$\mathcal{F}(\Theta)^\dagger_\rho \mathcal{F}(\Theta)_\rho = \frac{\beta_{\rho,\rho_0}}{\mathcal{F}(\Theta)^\dagger_{\rho_0}} \in \mathbb{C}^{D \times D}, \tag{17}$$

since we have already recovered the Fourier coefficient $\mathcal{F}(\Theta)_{\rho_0}$. The matrix $\mathcal{F}(\Theta)^\dagger_\rho \mathcal{F}(\Theta)_\rho$ is hermitian, and thus admits a square-root, that we denote $\mathcal{F}(\Theta)'_\rho$:

$$\mathcal{F}(\Theta)'_\rho = \left( \frac{\beta_{\rho,\rho_0}}{\mathcal{F}(\Theta)^\dagger_{\rho_0}} \right)^{1/2}. \tag{18}$$

The square-root $\mathcal{F}(\Theta)'_\rho$ only corresponds to $\mathcal{F}(\Theta)_\rho$ up to a matrix factor. This is an unidentifiability similar to the commutative case [28]. Specifically, consider the singular value decomposition (SVD) of $\mathcal{F}(\Theta)_\rho$:

$$\mathcal{F}(\Theta)_\rho = U.\Sigma.V^\dagger$$
$$\Rightarrow \mathcal{F}(\Theta)^\dagger_\rho \mathcal{F}(\Theta)_\rho = (U.\Sigma.V^\dagger)^\dagger.U.\Sigma.V^\dagger = V\Sigma^2 V^\dagger$$
$$\Rightarrow \mathcal{F}(\Theta)'_\rho = V\Sigma V^\dagger.$$

Thus, we have: $\mathcal{F}(\Theta)_\rho = UV^\dagger.\mathcal{F}(\Theta)'_\rho$ where $U, V$ are unitary matrices that come from the (unknown) SVD decomposition of the (unknown) $\mathcal{F}(\Theta)_\rho$. By recovering $\mathcal{F}(\Theta)_\rho$ as $\mathcal{F}(\Theta)'_\rho$, we fix $UV^\dagger = I$. This is similar to the same way in which [28] fixed $\phi = 0$ (rotation of angle 0, i.e., the identity) in the commutative case.

Next, we seek to find the remaining Fourier coefficients of the signal $\Theta$, from a limited subset of the bispectral coefficients. To this aim, we denote $\mathcal{R}$ the set of irreducible representations of the group

$G$. We recall that, for a finite group $G$, the set $\mathcal{R}$ is also finite, with its size equal to the number of conjugacy classes of $G$.

We consider the following bispectral coefficient:

$$\beta_{\rho_1,\rho_1} = (\mathcal{F}(\Theta)_{\rho_1} \otimes \mathcal{F}(\Theta)_{\rho_1})^\dagger C_{\rho_1\rho_1} \left[ \bigoplus_{\rho \in \rho_1 \otimes \rho_1} \mathcal{F}(\Theta)_\rho \right] C_{\rho_1\rho_1}^\dagger$$

$$= (\mathcal{F}(\Theta)_{\rho_1} \otimes \mathcal{F}(\Theta)_{\rho_1})^\dagger C_{\rho_1\rho_1} \left[ \bigoplus_{\rho \in \mathcal{R}} \mathcal{F}(\Theta)_\rho^{n_{\rho,\rho_1}} \right] C_{\rho_1\rho_1}^\dagger,$$

where $n_{\rho,\rho_1}$ is the multiplicity of irreps $\rho$ in the decomposition of $\rho_1, \rho_1$. This multiplicity is known as it only depends on the group, not on the signal $\Theta$.

We get an equation that allows us to recover additional Fourier coefficients of the signal $\Theta$:

$$\bigoplus_{\rho \in \mathcal{R}} \mathcal{F}(\Theta)_\rho^{n_{\rho,\rho_1}} = C_{\rho_1\rho_1}^{-1} (\mathcal{F}(\Theta)_{\rho_1} \otimes \mathcal{F}(\Theta)_{\rho_1})^{-\dagger} \beta_{\rho_1,\rho_1} C_{\rho_1\rho_1}^{-\dagger}, \qquad (19)$$

where everything on the right hand side is known. Therefore, every Fourier coefficient $\mathcal{F}(\Theta)_\rho$ that appears in the decomposition of $\rho_1 \otimes \rho_1$ into irreducible irreps $\rho$ can be computed, by reading off the elements of the block diagonal matrix defined by the direct sum. We recover the Fourier coefficients $\mathcal{F}(\Theta)_\rho$ for which $n_{\rho,\rho_1} \neq 0$.

We assume that this procedure provides at least one other Fourier coefficient, for an irreps $\rho_2$, that we fix. We can then compute the following bispectral coefficient:

$$\beta_{\rho_1,\rho_2} = (\mathcal{F}(\Theta)_{\rho_1} \otimes \mathcal{F}(\Theta)_{\rho_2})^\dagger C_{\rho_1\rho_2} \left[ \bigoplus_{\rho \in \rho_1 \otimes \rho_2} \mathcal{F}(\Theta)_\rho^{n_{\rho,\rho_2}} \right] C_{\rho_1\rho_2}^\dagger,$$

to get a novel equation:

$$\bigoplus_{\rho \in \mathcal{R}} \mathcal{F}(\Theta)_\rho^{n_{\rho,\rho_2}} = C_{\rho_1\rho_2}^{-1} (\mathcal{F}(\Theta)_{\rho_1} \otimes \mathcal{F}(\Theta)_{\rho_2})^{-\dagger} \beta_{\rho_1,\rho_2} C_{\rho_1\rho_2}^{-\dagger}, \qquad (20)$$

where everything on the right-hand side is known. Thus, every Fourier coefficient $\mathcal{F}(\Theta)_\rho$ that appears in the decomposition of $\rho_1 \otimes \rho_1$ into irreducible irreps can be recovered, by reading off the elements of the block diagonal matrix. We get the $\mathcal{F}(\Theta)_\rho$ for which $n_{\rho,\rho_2} \neq 0$. We iterate this procedure to recover more Fourier coefficients.

**Number of bispectral coefficients.**

Now, we show that our procedure can indeed recover all of the Fourier coefficients of the signal $\Theta$. Additionally, we show that it only requires a limited number $L$ of bispectral coefficients, where $L$ depends on the group $G$. Specifically, it depends on the Kronecker product table of $G$, which is the $|\mathcal{R}| \times |\mathcal{R}|$ table of the decomposition of the tensor product of two irreducible representations into a direct sum of elementary irreps. In this table, the element at row $i$ and column $j$ lists all of the multiplicity of the irreps that appear in the decomposition of $\rho_i \otimes \rho_j$.

The Kronecker table, i.e., any multiplicity $m_k = n_{\rho_k}$ of $\rho_k$ in the decomposition $\tilde{\rho} = \rho_i \otimes \rho_j$ can be computed with a procedure inspired by [9] and described below.

The procedure relies on the character $\chi_{\tilde{\rho}}(g) = \mathrm{Tr}(\tilde{\rho}(g))$ of the representation $\tilde{\rho}$ to be decomposed. From group theory, we know that the characters of irreps $\rho_i, \rho_j$ are orthogonal, in the following sense:

$$\langle \chi_{\rho_i}, \chi_{\rho_j} \rangle \equiv \frac{1}{|G|} \sum_{h \in G} \chi_{\rho_i}(h) \chi_{\rho_j}(h) = \delta_{ij}. \qquad (21)$$

Thus, we can obtain the multiplicity of irrep $\rho_k$ in $\tilde{\rho}$ by computing the inner product with the $k$-th character:

$$\langle \chi_{\tilde{\rho}}, \chi_{\rho_k} \rangle = \langle \chi_{\oplus_l m_l \rho_l}, \chi_{\rho_k} \rangle = \left\langle \sum_l m_l \chi_{\rho_l}, \chi_{\rho_k} \right\rangle = \sum_l m_l \langle \chi_{\rho_l}, \chi_{\rho_k} \rangle = m_k,$$

| $\otimes$ | $A_1$ | $A_2$ | $B_1$ | $B_2$ | $E$ |
|---|---|---|---|---|---|
| $A_1$ | $(1,0,0,0,0)$ | $(0,1,0,0,0)$ | $(0,0,1,0,0)$ | $(0,0,0,1,0)$ | $(0,0,0,0,1)$ |
| $A_2$ | $(0,1,0,0,0)$ | $(1,0,0,0,0)$ | $(0,0,0,1,0)$ | $(0,0,1,0,0)$ | $(0,0,0,0,1)$ |
| $B_1$ | $(0,0,1,0,0)$ | $(0,0,0,1,0)$ | $(1,0,0,0,0)$ | $(0,1,0,0,0)$ | $(0,0,0,0,1)$ |
| $B_2$ | $(0,0,0,1,0)$ | $(0,0,1,0,0)$ | $(0,1,0,0,0)$ | $(1,0,0,0,0)$ | $(0,0,0,0,1)$ |
| $E$ | $(0,0,0,0,1)$ | $(0,0,0,0,1)$ | $(0,0,0,0,1)$ | $(0,0,0,0,1)$ | $(1,1,1,1,0)$ |

**Table 2:** Kronecker table for the dihedral group $D_4$ which has 5 irreps called $A_1, A_2, B_1, B_2$ and $E$.

using the fact that the trace of a direct sum equals the sum of the traces (i.e. $\chi_{\rho \oplus \rho'} = \chi_\rho + \chi_{\rho'}$). Thus, we can determine the Kronecker product table of interest. For example, the Kronecker product table for the dihedral group $D_4$ is shown in Table 2.

The Kronecker product table shows us how many bispectral coefficients we need to complete our algorithmic procedure. Our procedure essentially uses a breadth-first search algorithm on the space of irreducible representations, starting with $\rho_1$ and using the tensor product with $\rho_1$ as the mechanism to explore the space. Whether this procedure succeeds in including all the irreps in $\mathcal{R}$ might on the group and its Kronecker (tensor) product table. Specifically, consider all the irreps that appear in the row corresponding to $\rho_1$ in the Kronecker product table. If these irreps do not cover the set $\mathcal{R}$ of all possible irreps, then the approach will need more than the tensor products of the form $\rho_1 \otimes \rho_j$ to get all of the Fourier coefficients of the signal $\Theta$.

$\square$

We observe in our experiments that this procedure does indeed succeed in computing all of the Fourier coefficients of the signal $\Theta$ for most of the groups of interest. We provide detailed examples of these computations on our github repository, for the diedral groups $D_4$, $D_{16}$ and for the chiral octahedral group $O$. The procedure does not succeed in the case of the full octahedral group $O_h$ does not succeed.

For the diedral group $D_4$, which has $\mathcal{R} = 5$ irreps, our approach allows us to recover a signal on $D_4$ from only 3 bispectral coefficients, instead of $5^2 = 25$. For the diedral group $D_{16}$, which has $\mathcal{R} = 11$ irreps, we recover the signal from 9 bispectral coefficients instead of $11^2 = 121$. For the octahedral group, which has $\mathcal{R} = 5$ irreps, we use 4 bispectral coefficients instead of $5^2 = 25$. This represent a substantial complexity reduction. More theoretical work is needed to prove that our approach applies to a wide range of discrete groups, or to further generalize it for groups such as $O_h$.

## G   Training and Implementation Details

The code to implement all models and experiments in this paper can be found at `github.com/sophiaas/gtc-invariance`.

For all experiments, we use near-identical, parameter-matched architectures in which only the type of invariant map differs. To isolate the effects of the invariant map, all models are comprised of a single $G$-Conv block followed by either Max $G$-Pooling or the $G$-TC layer, and an MLP Classifier. Here, we perform only *pure $G$-Conv*, without translation. Thus, we use filters the same size as the input in all models. The $G$-Conv block is comprised of a $G$-Conv layer, a batch norm layer, and an optional nonlinearity. For the Max $G$-Pool model, ReLU is used as the nonlinearity. Given the third-order nonlinearity of the $G$-TC, we omit the nonlinearity in the $G$-Conv block in the $G$-TC Model. The $G$-TC layer increases the dimensionality of the output of the $G$-Conv block; consequently the input dimension of the first layer of the MLP is larger and the weight matrix contains more parameters than for the Max $G$-Pool model. To compensate for this, we increase the dimension of the output of the first MLP layer in the Max Model, to match the overall number of parameters.

All models are trained with a cross-entropy loss, using the Adam optimizer, a learning rate of 0.00005, weight decay of 0.00001, betas of [0.9, 0.999], epsilon of $10^{-8}$, a reduce-on-plateau learning rate scheduler with a factor of 0.5, patience of 2 epochs, and a minimum learning rate of 0.0.0001. Each model is trained with four random seeds [0, 1, 2, 3], and results are averaged across seeds.

## G.1 MNIST Experiments: $SO(2)$ and $O(2)$ on $\mathbb{R}^2$

### G.1.1 Datasets

The $SO(2)$-MNIST dataset is generated by applying a random 2D planar rotation to each digit in the MNIST [36] training and test datasets. This results in training and test sets with the standard sizes of $60,000$ and $10,000$. For the $O(2)$-MNIST, each image is randomly flipped and rotated—i.e. transformed by a random element of the group $O(2)$. A random $20\%$ of the training dataset is set aside for model validation and is used to tune hyperparameters. The remaining $80\%$ is used for training. Images are additionally downsized with interpolation to $16 \times 16$ pixels.

### G.1.2 Models and Training

#### $C8$-CNN

**TC.** The $G$-Conv block consists of an $C8$-Conv block using 24 filters followed by Batch Norm. Next, the $G$-TC Layer is applied. The output is raveled and passed to a three-layer MLP using 1d Batch Norm and ELU nonlinearity and after each linear layer, with all three layers having output dimension $64$. Finally a linear layer is applied, with output dimension 10, for the 10 object categories.

**Max.** The $G$-Conv block consists of an $C8$-Conv block using 24 filters followed by Batch Norm and a ReLU nonlinearity. Next, a Max $G$-Pooling Layer is applied. The output is raveled and passed to a three-layer MLP using 1d Batch Norm and ELU nonlinearity and after each linear layer. The first layer has output dimension 275, to compensate for the difference in output size of the $G$-TC Layer. The remaining two layers having output dimension $64$. Finally a linear layer is applied, with output dimension 10, for the 10 object categories.

#### $D16$-CNN

**TC.** The $G$-Conv block consists of an $D16$-Conv block using 24 filters followed by Batch Norm. Next, the $G$-TC Layer is applied. The output is raveled and passed to a three-layer MLP using 1d Batch Norm and ELU nonlinearity and after each linear layer, with all three layers having output dimension $64$. Finally a linear layer is applied, with output dimension 10, for the 10 object categories.

**Max.** The $G$-Conv block consists of an $D16$-Conv block using 24 filters followed by Batch Norm and a ReLU nonlinearity. Next, a Max $G$-Pooling Layer is applied. The output is raveled and passed to a three-layer MLP using 1d Batch Norm and ELU nonlinearity and after each linear layer. The first layer has output dimension $2,380$, to compensate for the difference in output size of the $G$-TC Layer. The remaining two layers having output dimension $64$. Finally a linear layer is applied, with output dimension 10, for the 10 object categories.

## G.2 ModelNet10 Experiments: $O$ and $O_h$ acting on $\mathbb{R}^3$

### G.2.1 Datasets

The ModelNet10 dataset is downsampled and voxelized to a $10 \times 10 \times 10$ grid. for the $O$-ModelNet10 Dataset, each datapoint is transformed by a random cubic rotation. For the $O_h$-ModelNet10 Dataset, each datapoint is transformed by a random cubic rotation and flip. The standard training and testing sets are used. A random $20\%$ of the training dataset is set aside for model validation and is used to tune hyperparameters. The remaining $80\%$ is used for training.

### G.2.2 Models and Training

#### $O$-CNN

**TC**. The $G$-Conv block consists of an $O$-Conv block using 24 filters followed by a IID 3D Batch Norm Layer. Next, the $G$-TC Layer is applied. The output is raveled and passed to a three-layer MLP using 1d Batch Norm and ELU nonlinearity and after each linear layer, with all three layers having output dimension $64$. Finally a linear layer is applied, with output dimension 10, for the 10 object categories.

**Max.** The $G$-Conv block consists of an $O$-Conv block using 24 filters followed by a IID 3D Batch Norm Layer and a ReLU nonlinearity. Next, a Max $G$-Pooling Layer is applied. The output is raveled and passed to a three-layer MLP using 1d Batch Norm and ELU nonlinearity and after each linear

layer. The first layer has output dimension $5,420$, to compensate for the difference in output size of the $G$-TC Layer. The remaining two layers having output dimension $64$. Finally a linear layer is applied, with output dimension $10$, for the 10 object categories.

$O_h$-**CNN**

**TC.** The $G$-Conv block consists of an $O_h$-Conv block using 24 filters followed by a IID 3D Batch Norm Layer. Next, the $G$-TC Layer is applied. The output is raveled and passed to a three-layer MLP, with all three layers having output dimension $64$. Finally a linear layer is applied, with output dimension $10$, for the 10 object categories.

**Max.** The $G$-Conv block consists of an $O_h$-Conv block using 24 filters followed by a IID 3D Batch Norm Layer and a ReLU nonlinearity. Next, a Max $G$-Pooling Layer is applied. The output is raveled and passed to a three-layer MLP. The first layer has output dimension $20,000$, to compensate for the difference in output size of the $G$-TC Layer. The remaining two layers having output dimension $64$. Finally a linear layer is applied, with output dimension $10$, for the 10 object categories.

