# OpenReview forum: "A General Framework for Robust G-Invariance in G-Equivariant Networks"
_NeurIPS.cc/2023/Conference — NeurIPS 2023 poster_

### Official Review · Reviewer_aA5W · 2023-07-05

**Soundness:** 3 good
**Presentation:** 3 good
**Contribution:** 2 fair
**Rating:** 7
**Confidence:** 5

**Summary:**

The paper introduces a novel method to enhance robustness and group-invariance in group-equivariant convolutional neural networks (G-CNNs), named the G-triple correlation (G-TC) layer. This approach uses the concept of triple correlation on groups, a polynomial invariant map that is also complete, unlike many other invariant maps like the max. This "completeness" only removes variations caused by group actions but retains the signal's structure, contributing to the G-TC layer's strong robustness, particularly against invariance-based adversarial attacks. The authors also state that the G-TC layer results in improved classification accuracy over the standard Max G-Pooling in G-CNN architectures. An efficient implementation method for any discretized group is provided, and benefits are demonstrated on various groups and datasets.

The paper provides the context, highlighting the central role of the pooling operation in CNNs. The pooling operation has remained unchanged over the years, and while it serves its purpose of local-to-global coarse-graining of structure, it fails to construct robust representations that are invariant only to irrelevant visual changes. The paper then goes on to explain the role of group-equivariant convolutional networks (G-CNNs) and how they exploit group theory to achieve precise generalized group-equivariance. However, the pooling operation in G-CNNs is still susceptible to the lossiness of the max operation. To tackle this, the authors propose uncoupling the invariance and coarse-graining operations and introducing a new method for robust G-invariance through the group-invariant triple correlation. The authors demonstrate the superior performance of this approach through experiments showing improved classification accuracy.

**Strengths:**

The paper tackles the loss of information of the pooling operation in equivariant architectures and how to circumvent it. I really appreciated the fresh take on the importance of pooling and how even a very standard architectural block can still be improved. As a result, the paper brought to my attention the triple correlation operator which to my knowledge has yet to be extensively used in ML applications. Thus, this paper has a novel contribution that has the potential to benefit the community. Furthermore, I found the overall presentation and motivation of this paper exceptionally clear and thus it was an easy-to-read and parse paper. Finally, the empirical results do suggest that the introduced G-TC layer does have some benefits---although this analysis is a bit underdeveloped.

**Weaknesses:**

There are a few notable weaknesses of this paper. The first being that the proposed approach can only work on discrete groups due to the pairwise nature of the G-TC layer. Many of the current breeds of equivariant networks have found application domains with continuous groups such as $SO(3)$, $SE(3)$, etc ... but the current setup does not seem easy to extend here. Can the authors maybe comment on this or if this can be even a direction for future work?

Furthermore, the scaling of this is $O(|G|^2)$---I understand the reduction by a factor of 2---but this is still quadratic scaling. This means that many potential groups of potential interest are eliminated. For example, one finite group that is not considered in this paper but could possibly be used is the symmetric group of $N$ elements---i.e. $S_N$. But I believe the computational complexity here makes it not possible to scale this up to permutation networks that also have invariance. Can the authors comment on this?

The background section on signals should really be in the established language of fiber bundles. This has already been done multiple times in equivariant ML papers see E(2)-CNN, ESCNN, "A General Theory of Equivariant CNNs on Homogenous Spaces" Cohen et. al 2018.


The experiments in this paper are very limited to toy datasets in MNIST. I believe there is an avenue for multiple more important datasets that are larger scale. For example, E(2)-CNN has Cifar-10 but even this dataset is missing here. I encourage the authors to try larger-scale image datasets to show the benefits of their proposed approach.


**Minor**:
- Typo: Eqn 4. LHS your $g$ is written as $\tilde{g}$ but the paragraph below assumes regular $g$

**Questions:**

1. Please include a plot on runtime analysis for the training of a network of G-TC vs. a regular G-CNN with max pooling.

---

> ### Author Rebuttal · Authors · 2023-08-10
>
> Dear Reviewer aA5W,
>
> We thank you for the thoughtful review. We respond to your comments and questions below:
>
> **Discrete Groups**
>
> We thank the reviewer for mentioning the very interesting question of how to generalize our method to continuous groups. We first note that many of the papers that make use of continuous groups require choosing a discretization. In our paper, we apply our model to the cases of $SO(3)$ and $O(3)$ using the octahedral discretization of the groups, which is standardly used in many G-CNN papers.
>
> Steerable G-CNNs are the exception, as their theory applies to continuous groups. In the global response on Steerable G-CNNs, we show that the G-TC Layer can be naturally generalized to this setting. However, we note that the pairwise nature of the G-TC still requires us to discretize $G$. We have brainstormed a few ideas on how to overcome this. We emphasize that these ideas could also help reduce the computational cost of the G-TC layer, since the computation would no longer depend on the size of a discrete group.
>
> - Idea 1: Consider the (continuous) Lie group H. The G-TC on steerable functions leverages the action of $H$, we could think of considering instead the action of its Lie algebra $\mathfrak{h}$. We would then consider the generators of $\mathfrak{h}$ and show that computing the G-TC on the generators is enough. However, the complicated equation of the G-TC possibly makes this approach intractable.
>
> - Idea 2: We could take inspiration from steerable CNNs, which express any (learnable) equivariant convolutional filter bank as (learnable) coefficients on an equivariant basis computed offline. Similarly, we could think of expressing the G-TC as (fixed) coefficients on a basis of invariant operators. However, the convolution is a linear operator while the G-TC is not. Consequently, the linear algebra used for steerable CNNs do not apply.
>
> - Idea 3: Lastly, we could think about using implicit representations. Instead of computing the values of the G-TC on any possible pair $(g_1, g_2)$ we could instead represent it as a neural network, depending on $f$, that takes $(g_1, g_2)$ as input. However, it is unclear whether it would be a convenient data structure to work with. It is possible that the complexity cost of the TC would only be shifted into a computational cost of the NN.
>
> Although the generalization to the continuous case is not immediately obvious, we believe the above ideas could provide fruitful paths for exploration in future work. We believe, nonetheless, that presenting the theory for discrete groups is an important first step. In equivariant deep learning, the paper *Group-Equivariant Convolution Networks* (Cohen & Welling, 2016) introduced the G-CNN for discrete groups, and only later did the paper *Steerable CNNs* (Cohen & Welling 2017) present a theory that applies to continuous groups (while the authors still used the octahedral discretization in their experiments).
>
> **Computational Complexity \& Runtime Analysis**
>
> Please see the global response on computational complexity.
>
> **Fiber Bundle Formulation**
>
> We agree with the suggestion to present the background in the language of fiber bundles, as it is has become standard in the G-CNN literature. We did not do so in the first draft to avoid overcomplication for the reader, but think it will be worth it for consistency, and plan to make these edits in the final draft of the paper. Thank you for the suggestion!
>
> **Toy Datasets**
>
> We agree that the datasets we use here (MNIST and ModelNet10) are relatively simplistic. We made the decision to use simple datasets in our analyses because we can exactly control their mathematical structure. We generate our synthetic datasets by applying the actions of specific groups ($SO(2), O(2), SO(3), O(3)$) to these data exemplars. Thus, we know exactly the groups that structure the data, and can formulate theoretical expectations for the application of the G-TC layer. The CIFAR image dataset, by contrast, possesses many variations that are not attributable to group actions -- e.g. lighting changes, general differences in appearance across exemplars from the same category. Thus, theoretically, one expects less benefit from the use of group-invariant and group-equivariant structures with this dataset. Our goal in this work was to demonstrate empirically the theoretically expected properties of our proposed layer, which we think we have accomplished with the use of simple, controlled datasets.
>
>
> **Typo**
>
> Thanks for catching this! We’ve updated it in the paper.

---

> > ### Comment · Reviewer_aA5W · 2023-08-13
> > **Re: Rebuttal**
> >
> > I thank the authors for their detailed responses to my questions. The ideas about generalizing to Steerable CNNs are intriguing but still a bit raw. I am happy with most of their response and I will maintain my current score.
> >
> > Update:
> >
> > I've changed my mind after another read through the paper. I'm updating my score from 6->7 due to the fresh perspective on equivariance this paper brings.

---

> > > ### Author Response · Authors · 2023-08-18
> > > **Thank you!**
> > >
> > > Dear Reviewer aA5W,
> > >
> > > We thank you very much for your engagement with our rebuttal and appreciate the reconsideration of your score. Your review was very helpful for us and has already strengthened this work. We will gladly incorporate these clarifications into the final version of the paper.
> > >
> > > All the best,
> > >
> > > Submission8266 Authors

---

### Official Review · Reviewer_Axj4 · 2023-07-05

**Soundness:** 2 fair
**Presentation:** 2 fair
**Contribution:** 2 fair
**Rating:** 3
**Confidence:** 4

**Summary:**

This paper proposes a $G$-triple-correlation ($G$-TC) pooling layer in order to achieve group-invariance in group-equivariant convlutional neural networks (G-CNNs). Compared to the widely used $G$-pooling layers, the proposed $G$-TC layer is supposed to be "complete" in the sense that it removes only the variation due to the actions of the group, while preserving all information about the structure of the signal. In other words, the $G$-TC layer is injective over different group orbits. The authors claim that this property endows the $G$-TC layer with strong robustness, leading to resistance to invariance-based adversarial attacks.

**Strengths:**

1. The paper is relatively well written and well organized. It is easy to read.

**Weaknesses:**

1. The proposed method is only applicable to "regular" G-CNNs, where feature maps are signals over a group. It is hard to see how this can be extended to steerable G-CNNs where feature maps can be arbitrary fields.
2. One particular disadvantage of "regular" G-CNN is its computational and memory burden -- one has to physically store a function over the group $G$, which, after discretization, could be very large. The feature dimension after the proposed $G$-TC layer increases from $|G|$ to $|G|^2$. Even if there are potential ways to reduce the computational cost of the $G$-TC layer as speculated by the authors, this memory cost can not be circumvented.
3. The authors have claimed that the "completeness" of the $G$-TC layer leads to robustness to invariance-based adversarial attacks. However, there is neither theoretical nor empirical evidence to verify this claim.
4. Also, I am not completely convinced why "completeness" is so important. $G$-pooling is typically adopted close to the end of the model, by which time "coarse-graining" of the feature is usually intentional.
5. Also, can completeness be achieved by simple registration of the feature maps (e.g., according to the max-magnitude function value over the group)? In this case, the signal can stay at the length of $|G|$.
6. The experimental set up of this paper is non-canonical. There are too many adjustment to the baseline (e.g., same filter size as the input, only one $G$-Conv block, etc.) To make the results more convincing, the authors should simply replace the last $G$-max pooling layer of a E(2)-CNN with the proposed $G$-TC layer. I understand the authors are not trying to achieve SOTA, but the results displayed in Table 1 are simply too unconvincing.

**Questions:**

Please refer to the previous sectioin.

**Limitations:**

Yes.

---

> ### Author Rebuttal · Authors · 2023-08-10
>
> Dear Reviewer Axj4,
>
> Thank you for the time you’ve taken to read and review our paper. We address your comments and questions below:
>
> **Generalization to Steerable G-CNNs**
>
> Please see the global response on steerable G-CNNs.
>
> **Computational Complexity**
>
> Please see the global response on computational complexity.
>
> **Completeness & Robustness**
>
> *Theoretical evidence*. The completeness of the triple-correlation (TC) is well-established in the signal processing literature: It was first demonstrated by Yellott & Iverson (1992) for the case of the translation-invariant TC, and later generalized to the TC on compact groups by Kakarala (2009). We provide a statement of the completeness property in Section 3.2 Proposition 2, and point the reader to the original references for the proof, which is lengthy and requires substantial space and background to establish. As the G-TC Layer directly performs a triple correlation, it follows that the G-TC layer is also complete. Since the G-TC Layer is complete, it follows definitionally that it is robust to invariance-based adversarial attacks. In an invariance-based adversarial attack, the goal is to find inputs that are non-identical (up to group action) that yield identical output representations. Completeness states that the only inputs that yield identical outputs are identical up to group action. Thus completeness directly implies robustness to invariance-based adversarial attacks. Since this line of reasoning was not clear to the reviewer from our paper, we plan to make it much more explicit in the main body of the paper. We will also include an appendix that replicates the proof of completeness.
>
> We provide *empirical evidence* for the completeness of the G-TC layer in Section 5.2 and Figure 2, where we perform invariance-based adversarial attacks on the models trained in Section 5.1. We find empirically that our G-TC model is complete, whereas the Max G-Pool model is not. That is, when we optimize inputs to yield points identical to a target image in the output space of the G-TC model, every optimized input is identical to the target image up to group action. This analysis was first established by Sanborn, et al (ICLR 2023) and provides empirical demonstration that the theoretically-expected completeness property holds for the trained model. As this empirical demonstration of completeness is presented in the main paper along with a figure illustrating the results, we are not sure why it was missed by the reviewer. Do you have recommendations on how to improve its presentation?
>
> **The Importance of Completeness.**
>
> A key point that we emphasize in our paper is the distinction between *coarse-graining* and *invariance*. Standardly, pooling has been used in CNNs to accomplish both objectives. However, the concepts of coarse-graining and invariance are distinct and serve different purposes. We elaborate on this distinction in paragraphs 1- 4 of the introduction. In our paper, we introduce a layer that focuses on the problem of invariance – *not* coarse-graining. Spatial coarse-graining is still important, and standard pooling approaches can be used to achieve this. However, at the end of a G-CNN, the goal is to achieve invariance to the group action, to remove the G-equivariance preserved throughout the layers of the network and output a representation that is “canonicalized.” It is in this setting that completeness becomes important.
>
> The standard approach for achieving invariance at the output of a G-CNN is to perform Max G-Pooling, which returns the maximum output value over the group. This operation loses substantial information, as many non-identical inputs may yield identical max outputs. We provide an illustration of this problem in Figure 1 of the paper. Empirically we demonstrate that the lossiness of Max G-Pooling results in lower classification accuracy than the G-TC model when all else remains fixed. Thus, model performance reflects a tangible and substantial benefit of completeness.
>
> **Registration of Feature Maps.**
>
> The idea of registering the feature maps is an interesting alternative to the TC approach we take here. It sounds like your suggestion is something like: Take the maximum value of the output of the group, use that to perform registration – e.g. shift the outputs so that the max value appears in the first position of the vector (or something like this). This is an interesting idea worth exploring in future work. We foresee a few possible issues that may arise with this approach: First, if there are multiple equivalent (or similar) maximum values, there is an ambiguity as to how to register. In practice, due to variations in data, images from the same category that are not exactly identical up to transformation will yield different feature vectors and so the maximum is likely not a reliable guide to finding the canonical registration. Nevertheless, it would be worth experimenting to see if these issues matter in practice.
>
> Generally speaking, registration is a heuristic-based approach. Our work takes a mathematically-motivated approach that possesses theoretical guarantees. Future work may seek to test its limits: Are the theoretical gains worth the computational cost in comparison to a good-enough heuristic-based approach?
>
> **Non-Canonical Experimental Setup**
>
> In this work, we chose to test our new layer in what we see to be the simplest setting: a model possessing a single G-Conv layer with no translational convolution (only convolution on the group $G$). We agree it is a somewhat non-canonical setup. However, since we used an identical setting for both models and changed only the pooling method between them, we can still draw valid conclusions about the impact of that change. Our results show an unambiguous significant increase in accuracy with our pooling layer, across all datasets, averaged over many random seeds.

---

> > ### Comment · Reviewer_Axj4 · 2023-08-15
> >
> > I'd like to express my gratitude to the authors for their thorough response. I am particularly grateful for the comprehensive clarification on steerable networks. Nevertheless, I have lingering concerns regarding three areas:
> >
> > Completeness Consideration: I recognize that "excessive invariance," like the one introduced by max-G-pooling, might result in different group orbits providing the same response. Yet, is this truly a significant issue in applications such as image classification? After all, different group orbits representing the same digit should indeed be classified as that particular digit. There's a potential argument that striving for "completeness" may consume excessive memory resources to retain redundant data in the triple-correlation.
> >
> > Robustness Inquiry: The significance of Figure 2 in supporting the "completeness" assertion is evident to me. What remains unclear, however, is its association with "invariance-based adversarial robustness". My understanding suggests that an adversarially robust model should consistently produce comparable outputs for input group orbits that are closely related. This seems different from preventing distinct group orbits from generating identical outputs.
> >
> > Experimental Approach: I understand the rationale behind employing a non-canonical experimental setting to demonstrate the advantages of the proposed layer. However, to truly highlight its value in practical applications, wouldn't it be more compelling to conduct the experiments under standard conditions? Furthermore, in relation to my first point, as networks become deeper, could it be that the importance of "completeness" diminishes? It might be more desirable for different group orbits of the same digit to produce analogous outcomes.
> >
> > In conclusion, while I hold the authors' comprehensive response in high regard, I remain to be fully persuaded of the paper's overarching significance.

---

> > > ### Author Response · Authors · 2023-08-18
> > > **Completeness**
> > >
> > > We thank the reviewer for engaging with our rebuttal, and again express our appreciation for the interesting points brought up in the review, which have motivated the generalization to steerable networks and improved the quality of the work. We hope to answer remaining questions here, and we are happy to discuss any points further throughout the discussion period.
> > >
> > > **Completeness**
> > >
> > > Your question about completeness in the context of image classification highlights a distinction that we may have not made sufficiently clear. Thank you; we think this clarification will significantly strengthen the paper.
> > >
> > > We summarize your question as this: *“In image classification we want different examples of the same digit drawn in different styles to yield the same network output (the correct label). Completeness ensures that digits drawn in different styles will map to different outputs (orbit separation). Is this not too stringent for the problem of image classification? Are we wasting computational resources on the wrong problem?”*
> > >
> > > We highlight that there are two sources of variation that need to be eliminated in order to solve the Rotated MNIST task: (1) the action of the group (rotation); and (2) the style variation. The purpose of pooling over the group in G-CNNs (G-Pooling – Max or TC) is to remove the first kind of variation. The second kind of variation is removed by transforming the data into the hierarchical feature space defined by filters in each layer of the network. If the model is trained well, it will learn a good feature space that removes style variation.
> > >
> > > G-Pooling is applied at the end of the G-Conv network, after the image has been transformed through multiple G-Conv layers. G-Pooling operates on a *new transformed signal*–not the original image. Ideally, the network has learned a good set of features, so style is removed and all images of a particular digit class yield similar representations. However, since the network is G-equivariant, the rotation still remains (in a homomorphic sense). Thus, at this stage, we should (ideally) have ten “orbits”--one for each digit class. For each style-removed digit class, there is some vector that abstractly represents it. The orbit consists of all “rotated” versions of that vector–where "rotated" is in the sense of the group representation of SO(2) acting on the new feature space.
> > >
> > > Now, we want to remove the rotation and “canonicalize” the signal at the output of the network. One heuristic we can use is to take the maximum value of each filter output, since this will be invariant to rotation. This is Max G-Pooling. But, this lossy operation throws away information and destroys much of the network’s representation of the signal structure. Consequently, we can construct adversarial examples that yield the same network output and class label but look like *random noise*–shown in Figure 2 of the paper.
> > >
> > > The TC guarantees that we preserve *all* information about the signal, removing *only* the equivariant bit, thus “quotienting” out the group and achieving G-invariance in a completely lossless fashion, so that these kinds of adversarial examples cannot be created. This is shown in Figure 2 of the paper for the TC network. We show that the only inputs that yield identical outputs at this stage of the network are equivalent up to group action. This is far from a waste of resources; it results in concrete gains in classification accuracy (Table 1). Completeness is not “just” there to keep digit orbits separate. It is also there to *ensure robustness*, to prevent random noise patterns from being confused for digits and ensure that representations within the network are meaningful.
> > >
> > > So, to directly answer your question, we emphasize that the purpose of completeness here is to simply keep all of the information that is already in the last layer of the network, instead of throwing much of it away in a coarse and destructive manner via the max operation. It is not too stringent for the task of image classification; it does not require that different styles of the same digit remain separated in the output space. The network preceding the G-TC removes the style variation and collapses exemplars from the same class, leaving the G-TC layer the task of selectively removing the rotation without destroying the representation.

---

> > > > ### Author Response · Authors · 2023-08-18
> > > > **Experimental Approach**
> > > >
> > > > Now, we note that the “adversarial” images generated for the TC network in Figure 2 are all *exactly equal* to the target image (up to group action). Above, we said that the orbits at the output of the network should be digit *classes*, not digit exemplars. So, shouldn’t we expect the “adversarial” images to sample random images of the digit class in different orientations?
> > > >
> > > > In our **Experimental Approach**, we deliberately used only a single linear G-Conv layer to ensure that the network is unable to remove style variation, which is a nonlinear transformation (concretely, a subgroup of the diffeomorphic group). This allows us to disentangle the group-invariance of the TC from the style-invariance that is achieved by a deep network. Since we use only a single G-Conv layer, we force style variation to be removed entirely by the MLP that follows the G-Conv and G-TC. We can thus clearly evaluate the completeness of the G-TC layer with the analysis from Sanborn, et al (2023), with the expectation that the emerging “adversarial” examples will be exactly the target images in different orientations. We believe this is the cleanest and clearest way to demonstrate the expected theoretical completeness of the layer. With more G-Conv layers preceding the G-TC, we would expect to see with different styles of the same digit showing up in the “adversarial” images. While this would be interesting to observe, it would be a much less clear demonstration of completeness, as it would require the lengthy explanation given above for the reader to make sense of the conflation of style variation and group action.
> > > >
> > > > Nonetheless, this question about the relationship between style variation, group invariance, and completeness came up for you, so we believe it would be worthwhile to put a shortened version of the above explanation into the final version of the paper.

---

> > > > > ### Author Response · Authors · 2023-08-18
> > > > > **Robustness**
> > > > >
> > > > > In the robustness analysis, we focus on **Invariance-Based Adversarial Examples (IBAE)**, defined by [Jacobsen, et al (ICLR 2019)](https://arxiv.org/abs/1903.10484), because they can be used to test the completeness of the model. We copy the definition of IBAEs from the Jacobsen paper (Page 3, Definition 3):
> > > > >
> > > > > Let $\phi$ denote the $i$-th layer, logits or the classifier and let $x^* \neq x$ be in the $\phi$ pre-image of $x$ and and $o$ an oracle. Then, an invariance-based adversarial example fulfills $o(x) \neq o(x^* )$, while $\phi(x) = \phi(x^*)$ (and hence $D(x) = D(x^∗))$.
> > > > >
> > > > > Here,  $D$ is a classifier applied to the network output. In words, this means that $x^*$ and $x$ yield identical network outputs and receive the same class label, but are deemed are non-equivalent according to the oracle. To use IBAEs to evaluate completeness, we follow Sanborn, et al (2023). Here, the oracle is a function that determines whether $x$ and $x^*$ are in the same orbit.
> > > > >
> > > > > Importantly, this type of adversarial example is different from what you described above, which sounds more like Perturbation-Based Adversarial Examples (PBAEs, Definition 2 from Jacobsen et al (2019)). The idea behind PBAEs is essentially that the output should not “vary too much” when the input varies a little.
> > > > >
> > > > > We see your interest in PBAEs as related to the question about image classification: *"Similar but non-identical digits from the same class should yield similar representations. Does the TC ensure this?"*
> > > > >
> > > > > While we do not address PBAEs in our paper, we can demonstrate mathematically that the TC has this property. In particular, we show that the upper-bound on the (uniform norm of the) difference between two TCs is a value that depends on the (uniform norm of the) difference between the two signals, in the case of compact groups. Thus, small changes in the input yield small changes in the TC. We demonstrate this in a proof of the Lipschitz continuity of the TC.
> > > > >
> > > > > We encountered issues with LaTeX rendering in this response, so instead of writing it here, we provide a link to an image of the rendered TeX: https://postimg.cc/p9L3qWS0
> > > > >
> > > > > In short, the TC ensures that similar images (i.e. from the same class) should yield similar network outputs. But we note that the goal of our adversarial experiments was to demonstrate completeness, which we accomplish with IBAEs--a different kind of adversarial attack than what you describe in your comment (PBAEs)

---

> > > > > > ### Author Response · Authors · 2023-08-18
> > > > > > **Thank you!**
> > > > > >
> > > > > > Lastly, we want to thank you for your engagement during the discussion period. We hope that we were able to resolve your questions and provide more clarity on the motivation of our work. These discussions have been beneficial for us, and we would gladly incorporate these clarifications into the final version of the paper. If you have any remaining questions, we'd be happy to discuss further!
> > > > > >
> > > > > > All the best,
> > > > > >
> > > > > > Submission 8266 Authors

---

### Official Review · Reviewer_NbDh · 2023-07-06

**Soundness:** 4 excellent
**Presentation:** 3 good
**Contribution:** 3 good
**Rating:** 6
**Confidence:** 3

**Summary:**

This paper introduces a concept called G-triple-correlation (G-TC) layers, which are used in combination with G-equivariant convolutional layers to achieve G-invariant mapping functions. In contrast to conventional methods such as max or average G-poolings, which result in information loss, the proposed G-TC layers are completely identifiable: they selectively remove only the variations caused by group actions while preserving all the essential content information in images. The invariance and completeness of G-TC layers are proven under certain mild assumptions. The proposed G-TC layers are empirically validated for some representative examples, e.g., SO(2)-MNIST, O(2)-MNIST, SO(3)-ModelNet, and O(3)-ModelNet.



**Strengths:**

* The paper has a clear motivation, i.e., replacing the excessive invariance (G-pooling) with more informative and precise approach (G-TC).

* The proposed method is technically sound. The assumptions used in this paper seems to be mild. The useful theoretical properties (G-invariance, completness, uniqueness) and computational complexity of the proposed method are nicely presented.

* The paper is generally well-written and easy-to-follow. Figure 1 is a nice overview for the motivation of this paper.

* The experimental results are convincing, especially for the cases of 3-dimensional groups.

**Weaknesses:**

- As the authors mentioned, the main bottleneck of the proposed G-TC method is its computational cost.  Although the paper includes a theoretical analysis on the time complexity of G-TC, it would be beneficial if the study also presented experimental comparisons of the training and inference computational costs.

**Questions:**

- In line 302, the authors mention as "For the Max G-Pool model, ReLU is used as the nonlinearity. Given the third-order nonlinearity of the TC, we omit the nonlinearity in the G-Conv block in the TC Model." How would the performance be affected if ReLU is applied to G-Conv with G-TC?

- In line 303, "The *G-Pool* layer increases the dimensionality of the output of the G-Conv block". It seems that *G-TC* is correct?

**Limitations:**

Computational cost might be a bottleneck for the proposed G-TC method. This limitation is adequately addresed in the paper.

---

> ### Author Rebuttal · Authors · 2023-08-10
>
> Dear Reviewer NbDh,
>
> We thank you for the thoughtful review, and address your comments and questions below:
>
> **Computational Complexity**. Please see the global response on computational complexity.
>
> **ReLU & Nonlinearity**. For the G-TC model, we remove the ReLU that is standardly performed after the convolution because it *breaks completeness*. By pushing all negative values to zero, the ReLU operation eliminates information that the TC computation needs to retain completeness. Importantly, we find that removing the ReLU for the G-TC model doesn’t reduce classification performance. Rather, it improves it, due to the completeness property. The completeness be observed in the adversarial example experiments. For a G-TC model without ReLU nonlinearity, no out-of-orbit adversarial examples can be found (i.e. Figure 2 of the paper). By contrast, when ReLU is used directly prior to the G-TC layer, adversarial examples can be found – in line with theoretical expectations.
>
> **G-Pool Typo**. Thanks for catching this! We have updated it in the paper.

---

> > ### Author Response · Authors · 2023-08-18
> >
> > Dear Reviewer NbDh,
> >
> > As we near the end of the discussion period, we would like to ask if there are any remaining questions or concerns we could help resolve or clarify. Do let us know! We've appreciated your thoughts on our work and would be happy to discuss anything further.
> >
> > All the best,
> >
> > Submission8266 Authors

---

> > > ### Comment · Reviewer_NbDh · 2023-08-19
> > >
> > > Thank you for the comprehensive response. I believe this paper still deserves publication, so I am maintaining my review score as 6 and recommending its acceptance.

---

### Official Review · Reviewer_E2w2 · 2023-07-06

**Soundness:** 2 fair
**Presentation:** 3 good
**Contribution:** 2 fair
**Rating:** 5
**Confidence:** 4

**Summary:**

This work proposes to use the triple correlation to achive the group invariance. Unlike max pooling, the triple correlation preserves the entire information from the original signal, which may improve the quality of solutions for downstream tasks. Experiments show effectiveness of the proposed method.

**Strengths:**

1. The proposed method looks very natural. The proposed triple correlation layer achives the group invariance and at the same time preserves the entire information from the original signal. As far as I undertand from this paper, it seems to be the only group-invariant operation that enjoys this nice property in the deep learning literature.
2. The results from experiments look quite encouraging.

**Weaknesses:**

1. The novelty of the proposed method seem to be quite limited. It uses a well-known operation and apply it to NN.
2. The computational complexity of the proposed layer look quite high. Although the authors are able to save half of operations by using symmetry, the order of computation is still O(|G|^2).


**Questions:**

1. Since the triple correlation corresponds to bispectrum in frequency space, would it make more sense to optimize in Fourier space (or polar Fourier)?

2. Is the proposed method robust to noisy images? If I look at (8), multiplying three noisy terms may magnify the noise.

**Limitations:**

yes

---

> ### Author Rebuttal · Authors · 2023-08-10
>
> Dear Reviewer E2w2,
>
> Thank you for your review. We appreciate your attention and address your comments and questions below:
>
> **Novelty.** Although the triple-correlation in its translation-invariant form has a long history in signal processing, it is relatively unknown in deep learning. Google scholar reveals only a single paper (from the 90’s) that incorporates the triple-correlation (TC) into neural networks, in a very different context ([Delopoulos, et al, 1994](https://ieeexplore.ieee.org/abstract/document/286911?casa_token=6t3n5fTLMdEAAAAA:BpPufERn_ymV6UkxKtcd3GTb48yVSpfQtoP728NdQIAfcHCU3vOcC3YPxwgovRP3NhfkaO4YXA)). A handful of papers incorporate the bispectrum (BS) into deep learning, but largely as a pre-processing step rather than an architectural feature – with the exception of [Sanborn, et al, (ICLR 2023)](https://arxiv.org/abs/2209.03416). All of these papers use the generic translation-invariant form of the TC/BS – with the exception again of Sanborn, et al, (2023), which is the first to make use of the general group-invariant BS/TC applied to the problem of learning groups from data. Our work is the first to draw the connection between the group-invariant TC and the problem of *pooling and robustness in CNNs*. Here, we propose a novel computational primitive that exploits the structure of the G-convolution to construct invariant maps that significantly improve model accuracy and robustness over standard methods. Although the TC is not new, many great ideas have come from applying the right mathematics to a problem in computer science – such as the idea of a CNN itself :)
>
> **Computational Complexity.** Please see the global response regarding computational complexity.
>
> **Fourier Space.** Indeed, mathematically equivalent computations could be constructed in Fourier space using the bispectrum, the dual of the triple correlation in the frequency domain. The primary reason for performing the computation in the spatial domain is for compatibility with G-convolutional architectures, which live in the spatial domain. Spatial convolution is the dominant paradigm for computer vision, an only a handful of architectures perform equivalent computations in the frequency domain (e.g. Clebsch-Gordan Nets, Kondor et al, 2018). Nonetheless, our pooling method could be formulated in Fourier space for use in these architectures. In fact, Fourier space has some advantages with regard to computational complexity. In particular, for commutative groups, it permits an $O(|G|)$ reduction in the computation of the bispectrum (see [Kondor (2008) Group Theoretic Methods in Machine Learning, pg. 85](https://www.proquest.com/openview/d8ae7d9e47c87bb0acb05bf06ae8b6aa/1?pq-origsite=gscholar&cbl=18750)). For this paper, we chose the spatial formulation to maintain relevance to the existing literature. However, we believe the Fourier formulation is a promising avenue for future work.
>
> **Robustness to Noise.** Great question. The TC actually has special properties with respect to noise. In particular, the TC of a Gaussian signal is exactly zero. For this reason, it was originally used as a way to measure the “non-gaussianity” of a signal (See [Signal Processing with Higher-Order Spectra](https://ieeexplore.ieee.org/abstract/document/221324?casa_token=o59mtEpxSygAAAAA:TTSrggJvk8YjlWE_6UhREEMiZugsAm3OzD9OhCeyOTmp58nmk8-Lfn_zIaWpYScHKsjCzP7rNw) for more details). Thus, the TC has the nice property that it eliminates (Gaussian) noise. More generally, so long as the magnitude of the noise is $ < 1$, it will be reduced by the triple product rather than amplified. Keeping noise within this range can be encouraged with appropriate data normalization.
>
> **Questions for the Reviewer:**
> We note that you provided ratings of “2” (fair) for both soundness and contribution. Are there concrete results or analyses you like to see that would improve the quality of our paper in these categories?
>
> Again, we thank you for your time and attention! We look forward to discussing these topics further.

---

> > ### Comment · Reviewer_E2w2 · 2023-08-17
> >
> > Thanks for the detailed response, and it solves all my concerns. After reading all the comments and responses especially some major concerns from reviewer Axj4, I prefer not to change my score.

---

> > > ### Author Response · Authors · 2023-08-18
> > >
> > > Dear Reviewer E2w2,
> > >
> > > Thank you very much for reading our rebuttal. Your review and critical attention has already been helpful to our work. Regarding the concerns of Reviewer Axj4, we recommend that you take a look at our latest replies to them. We believe there were some confusions about completeness and the adversarial experiments that we were able to clarify. Axj4's questions indicate that these points were not clear enough in the original paper, so we intend to incorporate our responses into the final version, which will strengthen the paper considerably. We are thus quite grateful for this discourse. Please let us know if you have any further questions. We appreciate your engagement with our work!
> > >
> > > All the best,
> > >
> > > Submission8266 Authors

---

### Author Rebuttal · Authors · 2023-08-10

We thank the four reviewers for their time and attention. We have provided responses to each reviewer for unique points made in the review. Here, we highlight a few points that appear across reviews, to avoid redundancy.

**Generalization to Steerable G-CNNs**

We thank reviewer Axj4 for bringing up the very interesting idea of generalizing the G-TC to steerable G-CNNs, which emerges naturally from our framework -- as described here.

Consider a group $G$ that is the semi-direct product $G = \mathbb{Z}_2 \ltimes H$ of the group of translations $\mathbb{Z}_2$ and a group $H$ of transformations that fixes the origin $0 \in \mathbb{Z}_2$. Consider the feature map $\Theta: \mathbb{Z}_2  → \mathbb{R}^K$ that is the output of a steerable G-CNN. Here, $\Theta$ is a field that transforms according to a representation $\pi$ induced by a representation $\rho$ of $H$ on the fiber $R^K$ (Steerable CNNs. Cohen & Welling, 2017).

The G-TC can be applied to $\Theta$ by replacing the regular representation by the induced representation in Eq.(6). Specifically, replace any $\Theta(g)$, i.e. the scalar value of the feature map at group element g by $\pi(g)(\Theta)(x)$ i.e. the vector value of the feature map at $x$ after a group element $g$ has acted on it via the representation $\pi$:

$$T(\Theta) = \int_G  \pi(g)(\Theta)(x)^\dagger . \pi(g_1g)(\Theta)(x) . \pi(g_2g)(\Theta)(x) dg$$

Instead of computing the TC for each scalar coordinate k of $\Theta$, we directly compute it as a vector. The formulation does not depend on the choice of $x$ by homogeneity of the domain $\mathbb{Z}_2$ for the group $G$. Importantly, this TC is invariant to the action of the induced representation: the proof leverages the same arguments that the ones used in our Appendix. Consider a feature map $f_2 = \pi(g_0)[f]$ obtained from the action of $g_0$ on a feature map $f$, and show that $TC(f_2) = TC(f)$. The key ingredient of the proof is the change of variable within the integral $\int_G$, which follows the semi-direct product structure of G:

$h’ = h h_0$ and $t’ = \phi(h)t_0 + t$

where $\phi(h)$ is a matrix representing h that acts on $\mathbb{Z}_2$ via matrix multiplication: e.g. a rotation matrix $R = \phi(r)$ in the case of $SE(n)$. This concludes the proof.

In short, the definition of the G-TC and the proof of its invariance naturally emerge by replacing the regular representation by the induced representation. We thank the reviewer for this remark and would be happy to highlight this natural extension to steerable G-CNNs as an avenue for future work.

**Computational Cost**

Indeed, the TC has quadratic complexity that scales with the size of the group: $O(|G|^2)$. We acknowledge and describe this as a limitation in the paper. We note that it is quadratic in the size of the *group* -- not the size of the input.

Practical use of the TC can be limited to “small” groups and enjoy the benefits provided by our approach. For the groups of rotation, this means limiting to rotations discretized into multiples of $d$ degrees. This is the framework used in many equivariant neural networks (Group Convolution Neural Networks, Cohen & Welling 2016, experiments of Steerable CNNs, Cohen & Welling 2017, etc). For the groups of translations, this means limiting to images defined on a small grid, or to feature maps that have been coarse-grained and are now defined on a small grid. Coarse-graining is typically applied during earlier stages of the network, whereas the group-invariant G-TC layer is applied at the end of the network, at which point the signal is heavily coarse-grained and substantially reduced in size. Many groups of interest can discretized coarsely in G-CNNs without considerable loss of accuracy, and this makes the computational cost of the G-TC in practice less impactful.

This can be seen in the empirical time required for 1,000 training epochs for the G-TC and standard Max G-Pool model. For this analysis, we use the octahedral (rotational and full) models trained on the ModelNet10 dataset, averaged over all random seeds, and trained on a NVIDIA GeForce GTX 1080 Ti 12GB GPU. 1k epoch training times can be found below:

**Rotational Octahedral |G| = 24**

- Max G-Pool: 22.5 minutes
- G-TC: 37.5 minutes

**Full Octahedral |G| = 48**
- Max G-Pool: 29 minutes
- G-TC: 76 minutes

We note that the octahedral groups are relatively large for discretized groups used in practice. Still, the G-TC models only result in 1.67x and 2.62x increases in training time over the Max G-Pool models. We believe that there are many scenarios in which the strong improvements in classification accuracy (Table 1), and the completeness of the model (Figure 2) will be worth this increased cost.

One reviewer (aA5W) asked whether our approach could be applied to the symmetric group $S_n$ of permutations of $n$ elements, given its computational complexity. The size of the symmetric group itself is $n!$. This typically limits *any* computation on the full group, except in the regime of small $n$. Indeed, the TC would be faced with $O(n!^2)$ computations. Thus, we acknowledge that any exponentially-sized groups will pose a problem for the TC, except in the regime of small $n$. Nonetheless, many groups used in practice--such as the rotation, translation, and reflection groups--are continuous groups that possess natural discretizations that limit their size and thus the complexity of the TC in practice.

We would be happy to further emphasize this limitation in the paper, if the reviewers feel that it would help the readers and future users decide on how they can best integrate the G-TC in their applications. Additionally, we refer the reviewers to our answer to aA5W regarding the extension to continuous groups. There, we have included ideas that have the potential to reduce the computational cost in future work.

---

### Decision · Program_Chairs · 2023-09-21

**Decision:**

Accept (poster)

**Comment:**

The paper proposes a novel G-triple-correlation (G-TC) layer to achieve group-invariance in group-equivariant convolutional neural networks (G-CNNs), addressing the issue of information loss associated with traditional pooling operations. The method is technically sound and well-motivated, offering a fresh perspective on an established architectural component. While the reviewers commend the clarity of presentation and the innovative concept, they note limitations in computational cost as well as a need for further empirical validation in more benchmark datasets. Despite these weaknesses, the overall contribution is considered novel and potentially useful. Therefore, the paper is recommended for acceptance.